# The molecular mechanism of load adaptation by branched actin networks

Tai-De Li[1,2,3†], Peter Bieling[2,4,5*†], Julian Weichsel[6], R Dyche Mullins[4*], Daniel A Fletcher[1,2,7*]

[1]Department of Bioengineering & Biophysics Program, University of California, Berkeley, Berkeley, United States; [2]Division of Biological Systems & Engineering, Lawrence Berkeley National Laboratory, Berkeley, United States; [3]Advanced Science Research Center, City University of New York, New York, United States; [4]Department of Cellular and Molecular Pharmacology and Howard Hughes Medical Institute, University of California, San Francisco, San Francisco, United States; [5]Department of Systemic Cell Biology, Max Planck Institute of Molecular Physiology, Dortmund, Germany; [6]Department of Chemistry, University of California, Berkeley, Berkeley, United States; [7]Chan Zuckerberg Biohub, San Francisco, United States

**Abstract:** Branched actin networks are self-assembling molecular motors that move biological membranes and drive many important cellular processes, including phagocytosis, endocytosis, and pseudopod protrusion. When confronted with opposing forces, the growth rate of these networks slows and their density increases, but the stoichiometry of key components does not change. The molecular mechanisms governing this force response are not well understood, so we used single-molecule imaging and AFM cantilever deflection to measure how applied forces affect each step in branched actin network assembly. Although load forces are observed to increase the density of growing filaments, we find that they actually decrease the rate of filament nucleation due to inhibitory interactions between actin filament ends and nucleation promoting factors. The force-induced increase in network density turns out to result from an exponential drop in the rate constant that governs filament capping. The force dependence of filament capping matches that of filament elongation and can be explained by expanding Brownian Ratchet theory to cover both processes. We tested a key prediction of this expanded theory by measuring the force-dependent activity of engineered capping protein variants and found that increasing the size of the capping protein increases its sensitivity to applied forces. In summary, we find that Brownian Ratchets underlie not only the ability of growing actin filaments to generate force but also the ability of branched actin networks to adapt their architecture to changing loads.

*For correspondence:
peter.bieling@mpi-dortmund.
mpg.de (PB);
Dyche.Mullins@ucsf.edu
(RDycheM);
fletch@berkeley.edu (DAF)

†These authors contributed
equally to this work

Competing interest: The authors
declare that no competing
interests exist.

Reviewing Editor: Alphee
Michelot, Institut de Biologie du
Développement, France

## Editor's evaluation

Through the use of an elegant experimental setup, this study offers a molecular explanation for why branched actin filament networks, similar to those encountered in migrating cells, become denser when growing against a mechanical load. Importantly, the results also confirm the Brownian ratchet model for actin assembly. This study captures several important features of branched filament networks and should become a reference in the field.

## Introduction

Life relies on force. Every living cell must generate, resist, and/or transmit physical forces as it organizes its internal spaces and interacts with its external environment (*Janmey and McCulloch, 2007;*

*Kasza et al., 2007*). In eukaryotic cells, the forces required for many biological processes —such as cell division, amoeboid locomotion, and embryonic development— flow through the *actin cytoskeleton*, a set of polymer networks made of cross-linked and/or entangled actin filaments (*Blanchoin et al., 2014*; *Fletcher and Mullins, 2010*; *Pollard and Cooper, 2009*). Like the vertebrate skeleton for which it is named, the actin cytoskeleton alters its structure in response to applied forces. In vertebrate animals force causes trabecular bone to become denser and stronger, a response often called 'Wolff's Law' (*Wolff, 1892*). Some cytoskeletal structures, including *branched actin filament networks*, also respond to applied forces by becoming denser and stronger (*Bieling et al., 2016*; *Mueller et al., 2017*).

Branched actin networks are self-assembling cytoskeletal structures that harness the free energy of actin filament elongation to move and shape membranes in eukaryotic cells. Unlike myosin motor proteins, which are good at generating pulling (tensile) forces, branched actin networks excel at generating the pushing (compressive) forces required for many cellular processes, including protrusion of leading edge membranes in migrating cells (*Bisi et al., 2013*; *Wu et al., 2012*), motility of intracellular pathogens (*Welch and Way, 2013*), healing of cell ruptures (*Clark et al., 2009*), endocytosis (*Mooren et al., 2012*), phagocytosis (*Insall and Machesky, 2009*; *Jaumouillé et al., 2019*), and the formation of tight cell adhesions (*Yamaguchi et al., 2005*).

A membrane surface can locally direct formation of a branched actin network by promoting and coordinating interactions between five core components: (i) a nucleation promoting factor (NPF) related to the Wiskott-Aldrich Syndrome Protein (WASP), (ii) the Arp2/3 complex, (iii) capping protein (CP), (iv) filamentous actin, and (v) monomeric actin bound to profilin (*Achard et al., 2010*; *Akin and Mullins, 2008*; *Bieling et al., 2018*; *Loisel et al., 1999*). Branched actin network assembly begins when signaling molecules, such as Rho-family GTPases, cluster together and activate NPFs on a membrane surface (*Dominguez, 2009*; *Husson et al., 2010*). Active NPFs locally promote actin nucleation from the sides of pre-existing filaments by the Arp2/3 complex (*Mullins et al., 1998*; *Rohatgi et al., 1999*). These newly created actin filaments elongate at their fast-growing (barbed) ends from profilin-actin complexes (*Funk et al., 2019*), which are fed to the filaments by an intrinsic polymerase activity of the nucleation promoting factors (*Bieling et al., 2018*). Individual filaments elongate and push against the NPF-coated membrane surface, but only for a short time before capping protein terminates their growth (*Edwards et al., 2014*; *Schafer et al., 1996*). As a result, steady-state network growth requires continual nucleation. Whenever active NPF molecules are concentrated together on a membrane surface, this sequence of interactions is sufficient to create a powerful molecular motor capable of generating kilopascal (nN/μm$^2$) pressures (*Bieling et al., 2016*; *Parekh et al., 2005*; *Marcy et al., 2004*; *Wiesner et al., 2003*).

Each molecular motor exhibits its own, characteristic response to applied forces, and this response can provide insight into the motor's cellular function and underlying biophysical mechanism. Force, for example, coordinates the out-of-phase stepping of two-headed kinesins along a microtubule (*Yildiz et al., 2008*), and it can cause dynein motors to step 'backward', toward the microtubule plus end (*Gennerich et al., 2007*). Force also causes some myosin motors to cling more tightly to actin filaments (*Laakso et al., 2008*). Similarly, force produces dramatic effects on the motor activity of branched actin networks, increasing the number and density of actin filaments (*Bieling et al., 2016*; *Mueller et al., 2017*) as well as the mechanical efficiency of the network (*Bieling et al., 2016*). These changes in filament density also optimize the material properties of a branched actin network to better resist deformation and to generate higher forces, all without altering the stoichiometry of actin relative to other key network components—capping protein and the Arp2/3 complex. Load adaptation in a growing actin network comprises two distinct processes: (i) reorientation of filaments within the network (*Mueller et al., 2017*; *Weichsel and Schwarz, 2010*) and (ii) an increase in the steady-state number of growing filaments (*Bieling et al., 2016*). Filament reorientation has been explained by kinetic competition models (*Weichsel and Schwarz, 2010*; *Schaus et al., 2007*) but the mechanism underlying the force-induced increase in filament number remains unknown.

To figure out how compressive load forces increase the number of growing filaments, we studied branched actin networks assembled from purified proteins on micro-patterned, functionalized glass surfaces. We applied defined forces and measured the rate of network growth using a modified Atomic Force Microscope (AFM) cantilever and simultaneously visualized the flux of constituent molecules – actin, capping protein, and the Arp2/3 complex – into the network by total internal reflection

fluorescence (TIRF) microscopy (*Bieling et al., 2016*). From these measurements, we find that load adaptation in the network arises from a mismatch in the force-dependent activities of capping protein and the Arp2/3 complex. Contrary to our expectations, the overall rate of new branch formation is reduced by compressive forces. This reduction follows from inhibitory interactions between free barbed ends of actin filaments and surface-attached nucleation promoting factors, via processes we call 'monomer gating' (*Akin and Mullins, 2008*) and 'barbed-end interference' (*Funk et al., 2021*). Our key finding is that load forces cause the rate constant for filament capping to decrease exponentially and that this change accounts for the net increase in the density of growing filament ends with increasing force. Intriguingly, the force responses of filament elongation and capping turn out to be closely matched, ensuring that the average filament length and molecular stoichiometry of the network remain constant across a wide range of load forces. The exponential force response of the per-filament capping rate suggests that it is governed by a Brownian Ratchet similar to the one proposed to govern filament elongation against a load (*Hill and Kirschner, 1982*; *Peskin et al., 1993*; *Mogilner and Oster, 1996*; *Mogilner and Oster, 2003*). To test this idea directly we created 'bulky' capping protein mutants that add larger length increments to the barbed end of an actin filament. When added to freely growing actin filaments in solution, these bulky mutants exhibit no defects in capping activity, but when added to actin networks growing against an opposing force the mutants display dramatically different activity and cannot keep pace with wildtype capping protein. Overall, our work reveals that Brownian ratchets not only generate the force required to move membranes but also create tuned force responses that stabilize branched actin networks and enable them to respond to changing load forces.

## Results

### Steady-state assembly of actin networks from micro-patterned glass surfaces

To better understand load adaptation we used micropatterned glass coverslips to direct the assembly of polarized, branched actin networks, and then applied compressive forces to these growing networks with a calibrated atomic force microscope (AFM) cantilever (*Bieling et al., 2016*). Briefly, we mimicked the clustering of active nucleation promoting factors on a membrane surface by immobilizing the Arp2/3-activating, C-terminal region of WAVE1 (WAVE1ΔN) in micropatterned squares on the surface of glass coverslips (*Figure 1A*), functionalized with Poly-Ethylene Glycol-maleimide (*Fourniol et al., 2014*). Each coverslip contained 400 micropatterned squares of WAVE1ΔN, ranging in size from 1 μm x 1 μm to 50 μm x 50 μm. When we incubated these coverslips with purified protein components—stoichiometric profilin:actin complexes, the Arp2/3 complex, and capping protein (CP)—the WAVE1ΔN squares initiated assembly of polarized, branched actin networks. By confocal fluorescence microscopy, these networks formed three-dimensional 'pillars' growing from the coverslip surface at 7.3±1.6 μm/min —a rate comparable to actin assembly at the leading edge of migrating cells (*Fourniol et al., 2014*). As we showed previously (*Bieling et al., 2016*), network growth velocity in this system does not depend strongly on NPF pattern size; and the densities of network-associated actin, Arp2/3 and capping protein do not vary systematically from the outside edge of the pattern to the center (*Figure 1B*, *Figure 1—figure supplement 1*). These observations indicate that filament nucleation, elongation, and capping are not limited by the diffusion of soluble components through the network.

Once initiated, the actin networks rapidly settle into a phase of stable, steady-state growth that lasts for more than 1 hr (*Bieling et al., 2016*). The use of stoichiometric profilin:actin complexes and capping protein in our reaction mixtures strongly damps spontaneous filament assembly and creates a 'metastable' (*Pollard et al., 2000*) or 'dynamically stable' (*Pernier et al., 2016*) pool of monomeric actin that does not change significantly over the course of an experiment. Briefly, profilin forms a one-to-one complex with actin that damps spontaneous nucleation and prevents elongation of filaments from their pointed ends (*Pollard and Cooper, 1984*). Note also that our experiments are performed using non-muscle actin, which binds profilin with approximately 10-fold higher affinity than does the mammalian skeletal muscle actin used in many in vitro studies (*Bieling et al., 2018*; *Funk et al., 2019*). If a filament does form spontaneously, capping protein rapidly terminates growth from its barbed end and profilin-actin complexes cannot elongate its pointed end. Under these conditions, spontaneously

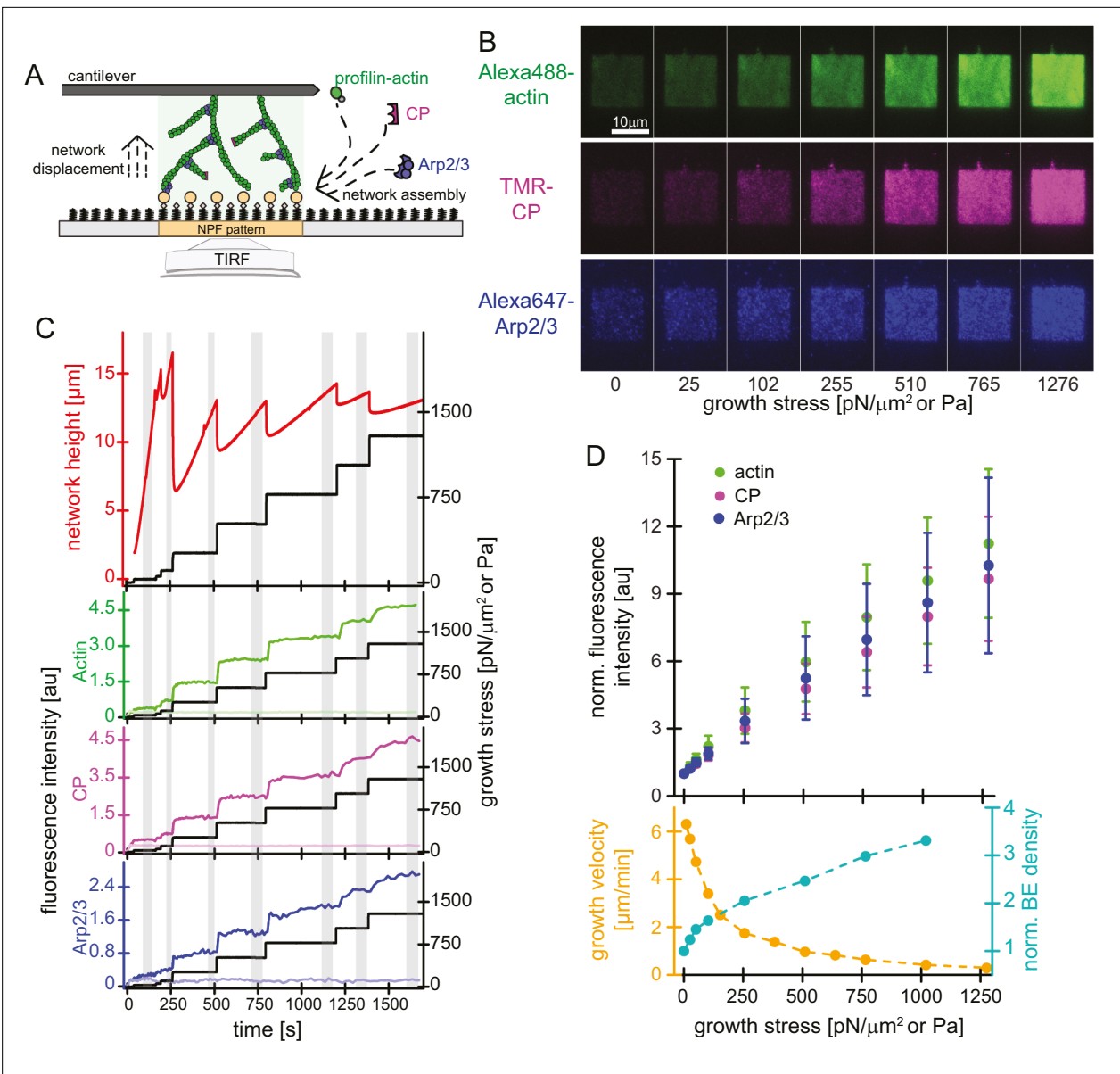

**Figure 1.** Effect of mechanical load on branched actin network assembly. (**A**) Schematic illustration of actin networks generated by profilin-actin, the Arp2/3 complex and capping protein from surfaces coated with NPF (WAVE1ΔN). Conditions are 5 µM actin (1% Alexa 488-labeled), 5 µM profilin, 100 nM Arp2/3 (5% Alexa647-labeled), 100 nM CP (15% TMR-labeled) if not indicated otherwise. (**B**) Representative TIRFM images of Alexa488-actin (top), TMR-CP (middle) and Alexa647-Arp2/3 (bottom) incorporation into dendritic actin networks at indicated growth stress. (**C**) Top: Height (red) and stress (black) as a function of time for a representative growing network. The stress was kept constant at a defined setpoint via the feedback mechanism of the AFM ("force-clamp mode") until the height change over time appeared linear and the network composition was constant, which means that a steady state in network assembly was reached. The stress was then raised to a higher setpoint, to which the network responded by a rapid adaptation, followed by a new steady state assembly phase. Network growth velocities and densities of components as a function of growth stress were determined from the linear, steady-state phases (grey areas). Bottom: Quantification of average fluorescent intensities for indicated protein components (left y-axis, colored lines) for networks either subjected to step-wise increasing loads (dark colors; applied stress is shown in the right y-axis (black lines)) or adjacent control networks growing in the same chamber in the absence of load (bright colors, see also *Figure 1—figure supplement 1*). (**D**) Top: Quantification of average fluorescent intensities for indicated network components as a function of applied load. Measurements are from for n=15 actin networks from N=5 independent experiments. Bottom: Corresponding average growth velocities (from n=12 actin networks from N=4 independent experiments) and average free barbed end densities (from n=21 actin networks; N=7 independent experiments) as measured by an 'arrest-and-label' approach as described in *Bieling et al., 2016*. Error bars represent ± SD (standard deviation).

The online version of this article includes the following source data and figure supplement(s) for figure 1:

**Source data 1.** Quantification of network height, growth stress, and fluorescence intensities of network components.

*Figure 1 continued on next page*

*Figure 1 continued*

**Figure supplement 1.** Uniform incorporation of all network components across the NPF surface.

**Figure supplement 1—source data 1.** Quantification of the spatial distribution of network components and time-dependence of their incorporation.

formed filaments are unstable and fall apart from their pointed ends (*DiNubile and Southwick, 1985*; *Young et al., 1990*; *Blanchoin et al., 2000*; *Pernier et al., 2016*). Only branched actin networks, formed by the micro-patterned nucleation promoting factors, can grow and survive in this mixture because their filaments are: (i) formed by continual nucleation at the coverslip surface and (ii) stabilized by Arp2/3 complexes bound to their pointed ends (*Blanchoin et al., 2000*).

Another reason the network growth rate is approximately constant for more than an hour is that only a small fraction of total protein in the reaction mix is consumed during this time. This can be demonstrated by a simple calculation. Each reaction contains 5 µM actin in a total volume of 150 µl. The branched actin networks growing from the micropatterned surface contain about 150 µM polymeric actin in the absence of compressive load (*Bieling et al., 2016*) in a maximum total volume of <0.01 µl (based on a total of 400 WAVE1ΔN squares with an average area of 50 µm$^2$, generating networks with a maximum height of <500 µm). The fraction of total actin used up during an hour, therefore, is less than 0.3%.

To quantify how the incorporation of actin, CP and Arp2/3 during network assembly adapts to load, we grew branched networks from squares of surface-attached WAVE1ΔN and used TIRF microscopy to visualize the density of all network components near the coverslip surface (*Figure 1A and B*). We used a calibrated AFM cantilever to apply a step-wise series of increasing load forces to a growing network, while simultaneously measuring network height and growth velocity (*Figure 1B and C*). Both the growth velocity and the density of all network components rapidly adapted to new growth forces and quickly established a new steady-state assembly rate, even at loads that nearly stalled network movement (*Figure 1B and C*). The growth velocity, determined from linear fits to the network height versus time during steady growth, fell sharply (*Figure 1C–D*, *Bieling et al., 2016*) whereas the steady-state density of network-incorporated actin, capping protein and Arp2/3 increased monotonically and very similarly with increasing load (*Figure 1B–D*). We confirmed that these abrupt changes in network density were not caused by changes in the concentrations of proteins in solution, because control networks assembling in the same chamber, not subjected to load showed a nearly constant density and composition over the duration of the experiment (*Figure 1—figure supplement 1*).

## The effects of load force on actin filament nucleation and capping

The steady-state nature of network assembly places strong constraints on our system. Specifically, at steady state the overall rates of nucleation and capping across the whole network must be equal ($R_{cap} = R_{nucleate}$). Otherwise, the filament density would not be constant but rather collapse or grow without bound. The overall nucleation rate is a complicated function that might depend on multiple factors, including the occupancy of the WH2 domains, the amount of surface-associated Arp2/3 complex, and the local density of polymeric actin. On the other hand, filament capping in our system appears to be a simple bimolecular interaction between soluble capping protein and free barbed ends. This is most easily demonstrated by the fact that the average filament length (i.e. the ratio of polymeric actin to either capping protein or the Arp2/3 complex within the network) varies as a simple inverse function of the capping protein concentration (*Wiesner et al., 2003*; *Akin and Mullins, 2008*). This means that the network-level rate of nucleation ($R_{nucleate}$, in units of sec$^{-1}$µm$^{-2}$) must equal the product of the soluble capping protein concentration ([CP], in µM), the surface density of free barbed ends (E, with units of µm$^{-2}$), and an appropriate capping rate constant ($k_{cap}$, with units of µM$^{-1}$sec$^{-1}$). In other words,

$$R_{cap} = k_{cap}[CP]\,E = R_{nucleate}$$

This expression can be rearranged to give the density of free barbed ends (*Equation 1*),

$$E = \frac{R_{nucleate}}{k_{cap}[CP]}$$

This key relationship (*Mullins et al., 2018*), imposed by the steady-state character of the system, means that any increase in density of free barbed ends must reflect *either* an increase in the rate of nucleation per unit area ($R_{nucleate}$) *or* a decrease in the per-filament capping rate ($k_{cap}$[CP]) *or* both.

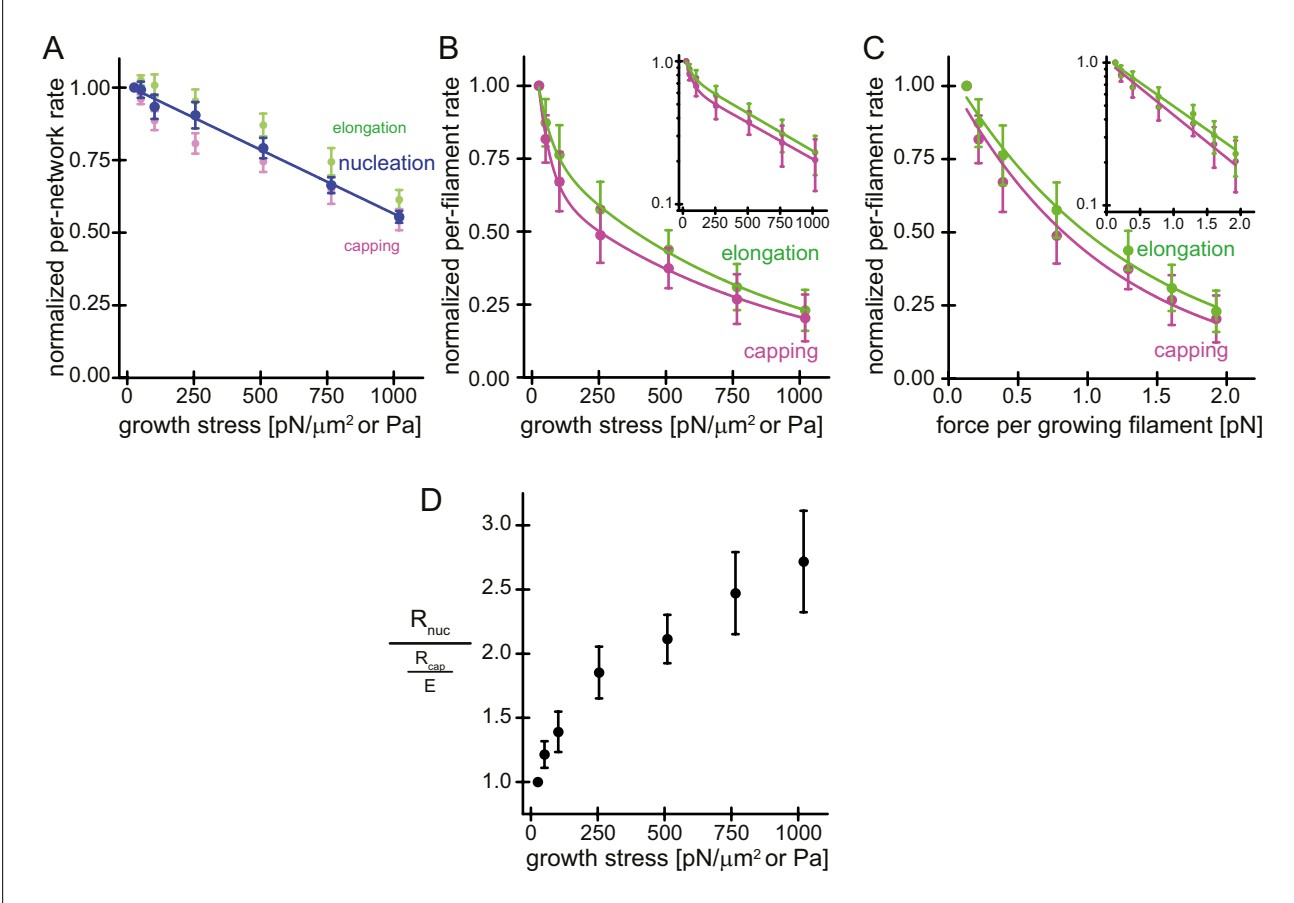

**Figure 2.** Load on branched actin network assembly decreases nucleation and. (**A**) Average per-network rates of filament elongation (green), capping (magenta), and nucleation (blue) calculated by the product of the bulk fluorescence intensities and the network growth velocity (*Figure 1D*) normalized to the flux at 25pN/μm2 as a function of external load. The line is a linear fit to the nucleation rate data. Error bars are SEM. (**B**) Average per-filament rates of filament elongation (green) and capping (magenta) as determined by normalizing their per-network rates (shown in A) by the relative density of free barbed ends (*Figure 1D*) as a function of external load. Lines are fits to double exponential decay functions. Inset: Semi-logarithmic plots of the same data. (**C**) Average per-filament rates of filament elongation (green) and capping (magenta) as a function of force per growing filament end. Lines are fits to single exponential decay functions. Inset: Semi-logarithmic plots of the same data. Error bars are SEM. (**D**) Ratio of the per-network nucleation and the per-filament capping rates as a function of external load. Error bars are ± SEM (standard error of the mean).

The online version of this article includes the following source data for figure 2:

**Source data 1.** Quantification of network incorporation rates of Actin, CP and Arp2/3.

Based on numerical simulations, *Carlsson, 2003* proposed that the rate of filament nucleation could increase under load as a result of the autocatalytic nature of Arp2/3 branching, but this has never been tested experimentally. Furthermore, as far as we can tell, the force-dependence of filament capping has never been investigated or incorporated into numerical simulations.

The network-level rates of actin filament nucleation, elongation, and capping can be calculated by multiplying the steady-state fluorescence intensity of labeled components in the TIRF field (*Figure 1D*) by the steady-state network growth velocity under each loading condition (*Figure 1D*). To our surprise, these network rates decreased linearly and very similarly with applied force for all components (*Figure 2A*), falling to approximately 50% of their initial values at load forces sufficient to stall network growth (~1200 Pa). Because the number of sites at which nucleation takes place (the number of surface-immobilized NPF molecules) remains constant and does not change over the course of our experiment, we can directly interpret the drop in the per-network nucleation rate as an effect on the nucleation process per se. This is quite different from the process of filament capping and elongation, because the number of sites at which these processes take place—the free barbed ends of actin filaments—changes with load (*Figure 1D* and *Bieling et al., 2016*). To compute an estimate for the

per-filament capping and elongation rates, we normalized the per-network capping and elongation rates (*Figure 2A*) by the free barbed end densities produced under the same loads (*Figure 1D*), which we previously measured using an 'arrest-and-label' method to visualize free filament ends in the network (*Bieling et al., 2016*). This analysis revealed a dramatic, force-induced reduction in the per-filament rates of elongation and capping, each of which can be fit by a double exponential decay (*Figure 2B*). When we also normalized load force to obtain the force per filament, however, the responses of both elongation and capping are well fit by a single exponential decay (*Figure 2C*), suggesting that both processes are governed by the same simple mechanism. The exponential drop in the per-filament elongation rate is consistent with Brownian Ratchet theories. As noted, filament capping is a bimolecular interaction whose rate is controlled by the concentration of capping protein ([CP]) and a second-order rate constant ($k_{cap}$). Applied force does not affect the total concentration of capping protein in the reaction, but does it decrease the *effective* concentration of capping protein at the coverslip surface by limiting its diffusion through the network? The answer to this question is 'no', based on the fact that we do not observe a gradient of capping protein incorporation, decreasing from the outside edge of the network to the center, under any of our load conditions (*Figure 1— figure supplement 1*). The effect of compressive loading must, therefore, be to decrease the rate constant for filament capping. We investigate this idea in more detail below.

In summary, we set out to determine the mechanism underlying force-dependent increases in filament density of branched actin networks by measuring both the overall nucleation rate and the per-filament capping rate. Both rates decrease under load but only the decrease in per-filament capping rate (driven by the force-sensitivity of the capping rate constant) can explain the increase in barbed end density (*Equation 1* and *Figure 2D*).

## The effect of load on Arp2/3 complex activity

Our observation that compressive forces decrease the overall rate of nucleation was surprising in light of previous theories (*Carlsson, 2003*), so we investigated the molecular mechanism responsible for this effect. We began by visualizing incorporation of individual Arp2/3 complexes into growing networks (*Figure 3A*) using TIRF microscopy. For these single-molecule experiments, we mixed trace amounts of fluorescent Arp2/3 with a large excess of unlabeled complexes (1:5000) and used this mixture to form branched actin networks from WAVE1ΔN-coated surfaces. At this low labeling ratio, individual fluorescent Arp2/3 complexes appear abruptly as diffraction-limited spots on the WAVE1ΔN-coated surface and then fade over time as they move with the growing actin network away from the coverslip surface and out of the TIRF illumination field (*Figure 3A*, *Video 1*). We identified network incorporation events as spots whose intensity decays exponentially with time, and we rejected fluorescent spots that disappear in a single step, due either to dissociation or photobleaching. Based on the measured Arp2/3 incorporation rate and the surface density of WAVE1ΔN on our coverslips (1850 μm⁻², *Bieling et al., 2018*), we determined a nucleation rate of 0.037 s⁻¹ per WAVE1ΔN molecule. This rate is surprisingly fast given previous measurements of the rate-limiting step of Arp2/3 activation in solution-based assays (*Helgeson and Nolen, 2013*; *Smith et al., 2013*; *Zalevsky et al., 2001*). Because release of the bound nucleation promoting factor from the nascent branch appears to limit nucleation in solution (*Helgeson and Nolen, 2013*; *Smith et al., 2013*), we speculate that retrograde forces generated by network growth may facilitate dissociation of the surface-bound NPF and accelerate nucleation in the context of a force-generating network.

Increasing load forces slowed the fluorescence decay of individual Arp2/3 molecules, because they slowed the network growth rate (*Figure 3A*, *Video 1*, *Bieling et al., 2016*). We calculated a time constant for each Arp2/3 incorporation event by fitting its fluorescence decay profile with a single exponential (*Figure 3C*). The distribution of these time constants for each applied load force followed a Gaussian distribution (*Figure 3D*) whose mean was inversely correlated with network growth velocity (*Figure 3E*). In these single-molecule measurements, the rate of Arp2/3 complex incorporation decreased only moderately (~20%) as the applied load increased from zero to near the stall force of the network (*Figure 3B*).

Both single-molecule and bulk fluorescence measurements revealed a decrease in the rate of Arp2/3 incorporation under load (*Figures 2A and 3B*), but the decrease in bulk fluorescence (~50%) was more pronounced than that measured for single molecules (~20%) over the same range of applied loads (0–1000 Pa). A possible explanation for this discrepancy could be that a significant fraction of

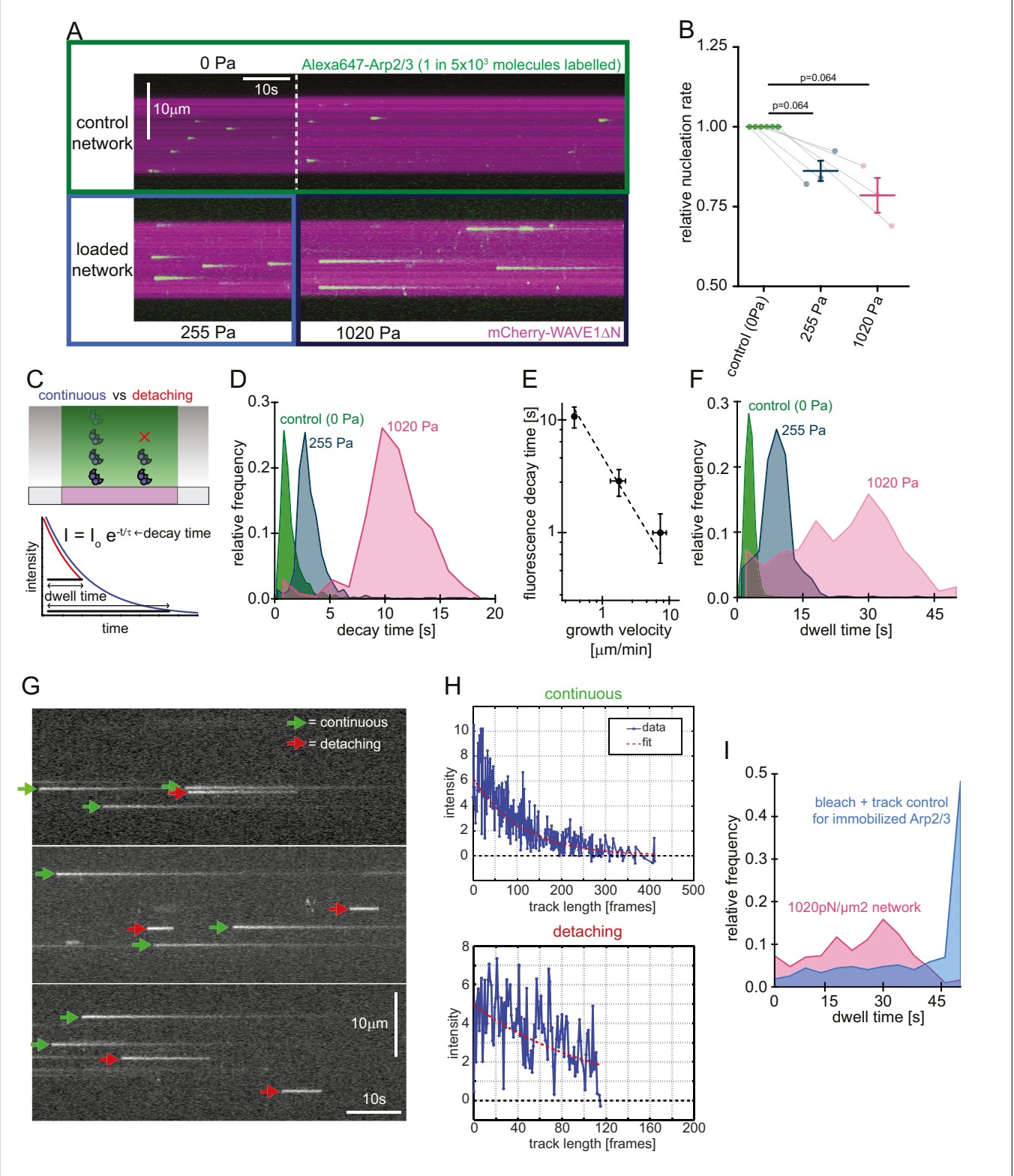

**Figure 3.** Single molecule characterization of force-dependent Arp2/3 nucleation. (**A**) Kymographs of single molecule nucleation on NPF surfaces (mCherry-WAVE1ΔN, magenta) by spike-in of a small fraction of Alexa647-Arp2/3 (green, c=20 pM) into the overall Arp2/3 pool (100 nM) at indicated applied stress (lower panel) or in an adjacent unloaded control network (upper panel). (**B**) Mean nucleation rates determined by single molecule imaging normalized to the nucleation rate in an adjacent unloaded control network at indicated growth stress from N=3 independent experiments. Pairs of control and loaded network are measured in the same flow chamber in a large field of view as illustrated by the lines linking two data points.

*Figure 3 continued on next page*

*Figure 3 continued*

Error bars are SEM. p-values were derived from paired Wilcoxon signed rank tests. (**C**) Scheme of single-molecule Arp2/3 dynamics followed by TIRF microscopy. Exponential decay of Arp2/3 fluorescence reflects movement of the fluorophore away from the coverslip driven by actin filament assembly. Some Arp2/3 molecules remain attached to the growing actin network for their entire transit through the TIRF illumination field (blue) whereas others detach from the network while still detectable by TIRF Illumination (red). Individual intensity trajectories, therefore, reflect either the transit time through the evanescent field or the time to detachment (dwell time). (**D**) Normalized frequency of fluorescence transit times of single Alexa 647-Arp2/3 complexes (n=1109, 594, and 318 Arp2/3 molecules at 0, 255, and 1020 Pa growth stress from N=3 independent experiments) in networks assembled at indicated stress. (**E**) Double-logarithmic plot of the mean fluorescence transit time (+/-SD) as a function of network growth velocity. The dashed line show perfect reciprocal correlation (slope = –1). (**F**) Normalized frequency of fluorescence dwell times of single Alexa 647-Arp2/3 complexes (n=1109, 594, and 318 Arp2/3 complexes at 0, 255, and 1020 Pa growth stress from N=3 independent experiments) in networks assembled at indicated stress. (**G**) Examples of kymographs from TIRF microscopy of individual Arp2/3 complexes in networks under high load (1020 pN/m2). Individual complexes are either continuously moving towards the rear of the evanescent field (continuous, green arrows) or dissociating prematurely (abortive, red arrows). (**H**) Representative time courses of fluorescence intensity for individual Arp2/3 complexes as a function of number of imaging frames categorized as either continuous (top panel) or abortive (bottom panel). (**I**) Relative frequency of dwell times for Arp2/3 complexes in dendritic networks at high load (1020pN/m2, red, n=318 Arp2/3 molecules from N=3 independent experiments) compared to the bleaching and loss of tracking control for surface-immobilized Arp2/3 complexes (blue, see *Figure 3—figure supplement 1*, n=274 Arp2/3 complexes from N=3 independent experiments). Note that the frequency of early loss events is exceeding the combined bleaching and tracking loss frequency.

The online version of this article includes the following source data and figure supplement(s) for figure 3:

**Source data 1.** Quantification of branching nucleation from single molecule Arp2/3 imaging.

**Figure supplement 1.** Tracking and bleaching control for single surface-immobilized Arp2/3 complexes.

**Figure supplement 1—source data 1.** Comparison of biochemical activities, effects on actin network architecture, and diffusion of wildtype and biulky mutant capping proteins.

---

the Arp2/3 complexes counted in the single molecule assay rapidly dissociate from the network and would, therefore, not contribute much signal to the bulk fluorescence assay. To test this idea, we measured the time between appearance and disappearance (dwell time) of each Arp2/3 molecule that we classified as incorporating into the network (*Figure 3F*, Materials and methods). Under low load (0 Pa), the distribution of Arp2/3 dwell times is well approximated by a single Gaussian peak, and the mean dwell time increases with decreasing network velocity (*Figure 2F*). At intermediate loading (255 Pa) and especially under high loads (1020 Pa), however, the distribution develops a 'shoulder' of dwell times that are shorter than expected for molecules that are simply moving away with the growing actin network from the coverslip and out of the TIRF illumination field (*Figure 3F*). Fluorescence decay curves of these short-lived events resemble truncated exponentials, consistent with an initial phase of linear motion away from the coverslip followed by abrupt dissociation of the Arp2/3 complex from the network (*Figure 3G–H*). These events were too frequent to be accounted for by experimental artifacts such as photobleaching or tracking glitches (*Figure 3I*, *Figure 3—figure supplement 1*). When we account for these rapid dissociation events, the single-molecule data agrees with bulk florescence measurements. Away from the coverslip, the Arp2/3 complex remains stably associated with the network, so rapid dissociation appears to be limited to a narrow zone very close to the coverslip surface. We speculate that the population of rapidly dissociating Arp2/3 reflects force-induced failure of newly formed branches, which appear to be particularly vulnerable.

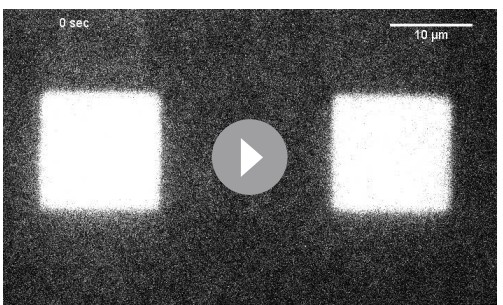

**Video 1.** Incorporation of individual Arp2/3 complexes into growing actin networks visualized by single-molecule TIRFM. Time-lapse movie from TIRF microscopy of single Alexa647-Arp2/3 complexes incorporating in actin networks either subjected to 1020 Pa of load (left square) or growing in the absence of external load (right square). Spike in of low amounts of Alexa647-Arp2/3 complexes (1–5000 labeling ratio, 20pM of 100 nM total) allows for the visualization of nucleation with single molecule resolution. Productive events are characterized by a rapid appearance of a fluorescent spot that decreases gradually in intensity as the Arp2/3 complex moves out of the evanescent field with the growing actin network. Scale bar = 10 μm. Conditions are as in Figure 3A.

https://elifesciences.org/articles/73145/figures#video1

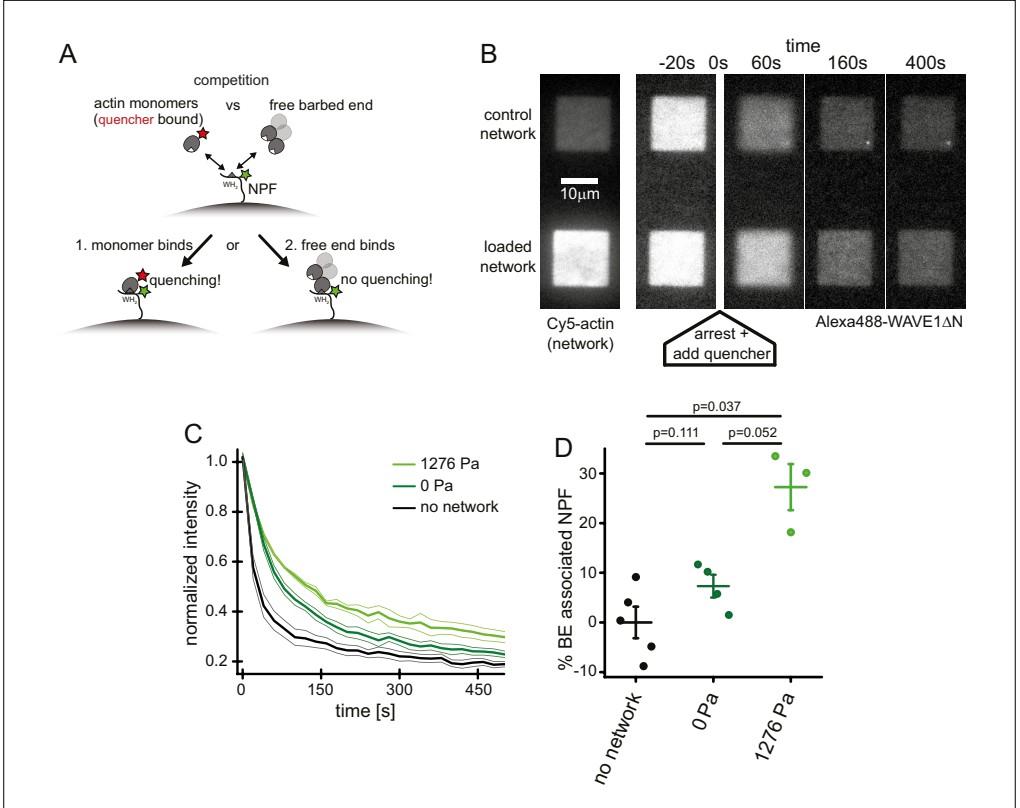

**Figure 4.** Free barbed ends bind and sequester the WH2 domain of the NPF in a load-dependent manner. (**A**) Scheme of the FRET setup. Surface-bound, donor- (Alexa488-) labeled NPF molecules can interact with either quencher- (Atto540Q-) labeled actin monomers resulting in decrease of donor fluorescence or unlabeled terminal protomers of uncapped barbed ends resulting in no change in fluorescence. The terminal protomers are unlabeled since quencher-labeled monomers are introduced only upon network arrest. (**B**) Time lapse TIRF microscopy images of Alexa 647-Actin (left image) or Alexa 488-WAVEΔN (FRET donor, other images) at indicated times after addition of 200 µl fixation and quenching mix (t=0, 30 µM LatB, 30 µM phalloidin, 5 µM Atto540Q-actin, 7 µM profilin, 37.5 µM myotrophin/V1 (CP inhibitor)) to 100 µl network assembly mix. (**C**) Averaged time-courses of the Alexa 488-WAVEΔN signal following the addition of quencher-labelled monomers at t=0 as shown in B for N=5, 4, and 3 experiments for no network, 0 Pa or 1278 Pa growth stress networks. Error indicators are SEM. (**D**) Mean fraction of barbed end-associated NPF molecules in either in the absence of an actin network (black) or in the presence of a non-loaded (dark green) or 1276 Pa loaded (light green) network (see Materials and methods). N=5, 4, and 3 experiments for no network, 0 Pa or 1278 Pa growth stress networks. Error bars are SEM. p-Values were derived from Mann–Whitney U tests.

The online version of this article includes the following source data and figure supplement(s) for figure 4:

**Source data 1.** Quantification of actin monomer binding by surface-immobilized WAVE1ΔN molecules as measured by FRET.

**Figure supplement 1.** Comparison of biochemical activities, effects on actin network architecture, and diffusion of wildtype and biulky mutant capping proteins.

## Effect of load on actin-WH2 interactions

Why do compressive forces reduce the rate of Arp2/3-dependent nucleation? In addition to promoting nucleation by delivering actin monomers to the Arp2/3 complex, WASP-family proteins associate with actin filament barbed ends via their WH2 domain and thus tether dendritic networks to the surfaces they push against (*Co et al., 2007*; *Funk et al., 2021*). This means that free barbed ends and actin monomers directly compete to occupy available NPF WH2 domains (*Figure 4A*). Because force increases the concentration of free barbed ends near the WAVE1ΔN-coated surface (*Figure 1D*, *Bieling et al., 2016*), we wondered whether these additional filaments inhibit nucleation by interfering with the ability WAVE1ΔN to bind monomeric actin via its WH2 domain. We adapted a recently developed

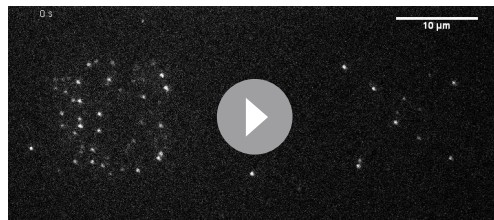

**Video 2.** Compressive load inhibits actin monomer binding to nucleation promoting factors. Time-lapse movie from TIRF microscopy of two WAVE squares containing Alexa 488-WAVEΔN (FRET donor attached to the NPF) generating actin networks either subjected to 1020 Pa of load (left square) or growing in the absence of external load (right square). At the indicated time (t=60 s), 200 μl fixation and quenching mix 30 μM LatB, 30 μM phalloidin, 5 μM Atto 540-actin, 7 μM profilin, 37.5 μM Myotrophin (CP inhibitor) were added to 100 μl network assembly mix, leading to the observed drop in donor fluorescence. Scale bar = 10 μm. Conditions are as in Figure 4B.
https://elifesciences.org/articles/73145/figures#video2

Förster Resonance Energy Transfer (FRET) assay (*Bieling et al., 2018*) to directly measure partitioning of WH2 domains between actin monomers and filament barbed ends (*Figure 4A*). Briefly, we labeled a WH2-adjacent site of WAVE1ΔN with a fluorescent donor (Alexa488) and conjugated a non-fluorescent acceptor (Atto540Q) to monomeric actin using a labeling protocol that does not perturb binding to profilin or WH2 domains (*Bieling et al., 2018*; *Funk et al., 2021*).

To measure partitioning of WH2 domains between actin monomers and filament barbed ends we micro-patterned glass coverslips with WAVE1ΔN, doped with 30% donor-labeled molecules. We then assembled dendritic networks under zero or high load (1276 Pa) in the absence of quencher-labeled actin (*Figure 4B*). We simultaneously arrested network growth and inhibited filament capping by adding a combination of soluble factors: (1) Latrunculin B, to bind monomeric actin and prevent polymerization; (2) Phalloidin, to stabilize existing filaments and prevent network disassembly; and (3) myotrophin/V-1, to bind and inhibit soluble capping protein. This combination of inhibitors arrests network growth while preserving free barbed ends (*Bieling et al., 2016*). At the same time, we also added quencher-labeled actin monomers that were pre-incubated with Latrunculin B to prevent their incorporation into free filament ends. The quencher-labeled monomers induced a rapid drop in donor fluorescence (*Figure 4B–C* and *Video 2*) as they bound available WH2 domains. We interpret donor fluorescence remaining at long time scales to reflect WH2 domains protected from quencher-labeled monomers by interaction with filament barbed ends.

To quantify the fraction of WH2 domains bound to free barbed ends, we compared the residual donor fluorescence of WAVE1ΔN in the absence and presence of a dendritic actin network (*Figure 4C*). In unloaded networks, approximately 7% of WH2 domains are protected from quenching, while under high load the protected fraction increases by ~3.7-fold, to 27% (*Figure 4D*). These numbers are in striking agreement with the 3.3-fold increase in free barbed end density observed in networks under similar load forces (*Figure 1D*, *Bieling et al., 2016*). The 20% decrease in available WH2 domains also agrees well with the load-induced 20% drop in overall nucleation rate (*Figure 3B*). In summary, these results verify that applied forces raise the number of free barbed ends that in turn engage an increasing number of nucleation promoting factors in non-nucleating complexes (*Funk et al., 2021*; *Mullins et al., 2018*). This type of negative feedback mechanism, which we call 'barbed end interference' (*Funk et al., 2021*), quantitatively explains the force-induced decrease in Arp2/3-dependent nucleation.

## Brownian Ratchet theory explains the kinetics of filament capping under load

As described above (*Figure 2B–C*), actin filament elongation and capping follow almost identical force response curves, which explains our previous observation that the average filament length in a branched actin network does not change with applied load (*Bieling et al., 2016*). If the two responses were not matched, filaments would become longer or shorter with applied load, depending on whether the rate of capping was more or less sensitive to force. We confirmed that increasing the concentration of capping protein decreases the average filament length in a growing network, resulting in sparser networks of shorter filaments that grow faster under comparable loads (*Figure 5—figure supplement 1*). In line with our previous work (*Akin and Mullins, 2008*), we observed that the increase in capping was compensated by elevated nucleation rates, confirming that capping protein stimulates nucleation in branched networks (*Figure 5—figure supplement 1*, *Akin and Mullins, 2008*; *Funk et al., 2021*).

Remarkably, however, filament length remained nearly insensitive to applied load at each concentration of capping protein we tested (*Figure 5A*).

Although never explicitly predicted, the matched force responses of the per-filament capping and elongation rates are implicit in all Brownian Ratchet theories of force generation (*Mogilner and Oster, 1996*; *Peskin et al., 1993*). According to these theories, the rate at which a protein binds the end of a filament that is growing against a boundary depends on how often thermal motion opens a large enough gap to accommodate the incoming protein (*Figure 5B*). Intriguingly, the atomic structures of monomeric actin and heterdimeric capping protein (*Funk et al., 2021*; *Kim et al., 2010*; *Narita et al., 2006*) reveal that both require a 2.7 nm gap to bind the barbed end of an actin filament (*Figure 5C*, top and middle). Therefore, according to Brownian Ratchet theory, altering the size of capping protein should alter the force response of filament capping and break the lock-step that it normally keeps with filament elongation.

To test this prediction we constructed a 'bulky' capping protein mutant by fusing glutathione S-transferase (GST), which forms homodimers, to the C-termini of both α and β subunits of capping protein (*Figure 5C*, bottom). From existing structures, we predict that the gap required for this bulky variant to bind to the filament end should be significantly larger than for wildtype capping protein (*Funk et al., 2021*; *Kim et al., 2010*; *Narita et al., 2006*). Under load, these larger gaps open much less frequently, and so the rate of capping by the bulky variant should decrease much more strongly with applied force. We confirmed that the bulky capping protein variant caps filament barbed ends with wildtype kinetics in solution (*Figure 5—figure supplement 2*). To directly compare the force sensitivities of wildtype and bulky capping protein under the same conditions, we constructed dendritic actin networks using mostly (90%) unlabeled wildtype capping protein, doped with small amounts (5% each) of wildtype and bulky capping proteins labeled with different fluorescent dyes. We labeled wildtype capping protein with tetramethyl-Rhodamine (TMR) and the bulky variant with Alexa-647. Under low load forces both wildtype and bulky capping proteins incorporated into the network, but only wildtype capping protein followed the same force response as actin and maintained a constant stoichiometry with actin under all loading conditions (*Figure 5D*). In contrast, the ratio of the bulky capping protein mutant to actin in the network diverged under load (*Figure 5D–F*, *Video 3*). To determine whether this effect was caused by decreased diffusion of the bulky capping protein through the denser actin networks produced under high load, we used line scans to quantify the distribution of capping protein from the outer edge of the WAVE1ΔN squares to the center. This analysis revealed no spatial variation consistent with decreased diffusion of the bulky capping protein mutant into the network (*Figure 5—figure supplement 2*). We also tested for tag-specific effects by using a bulky NusA-tag instead of a GST homodimer at the N-terminus of capping protein, which resulted in a similar force-dependent reduction of incorporation (*Figure 5—figure supplement 2*).

To test whether the altered force-response of the bulky capping protein can be quantitatively described by Brownian Ratchet theory, we developed an analytic model of actin network growth under load (see Appendix 1). Briefly, we assumed that the rate constant for capping declines exponentially with opposing force with $k_{cap} \sim e^{\frac{-f \cdot \delta \cdot sin\theta}{K_B T}}$ , where the $K_B$ is the Boltzmann constant, $T$ is the temperature, $\theta$ is the contact angle of filaments to the encounter surface, $f$ is the force of one individual filament against the surface, and δ is the gap size required for incorporation of capping protein. The latter parameter differs for wildtype capping protein and its bulky variant, leading to their differential response to load. The average filament contact angle changes in response to force (*Bieling et al., 2016*; *Mueller et al., 2017*; *Weichsel and Schwarz, 2010*) and was estimated from the actin density and number of free barbed ends at various forces as measured previously (*Bieling et al., 2016*). To obtain the force $f$ that opposes a single polymerizing filament, we divided the total external force on the whole network by the measured force-dependent number of free barbed ends sharing this load (*Figure 1C*; *Figure 1—figure supplement 1*; *Bieling et al., 2016*). Finally, we took into account the internal tethering (equivalent to a frictional force opposing motion) that arises from the interaction between free barbed ends and NPF proteins on the surface (*Figure 4*). Except for this characteristic tethering force, all parameters are derived from experimental data. Remarkably, even with all but this single free parameter constrained, the model accurately matches the force-dependent change in the relative incorporation of wildtype capping proteins over its bulky variant (*Figure 5E*). Furthermore, the characteristic tethering force yielded from this model (0.3 pN) would be consistent with a weak

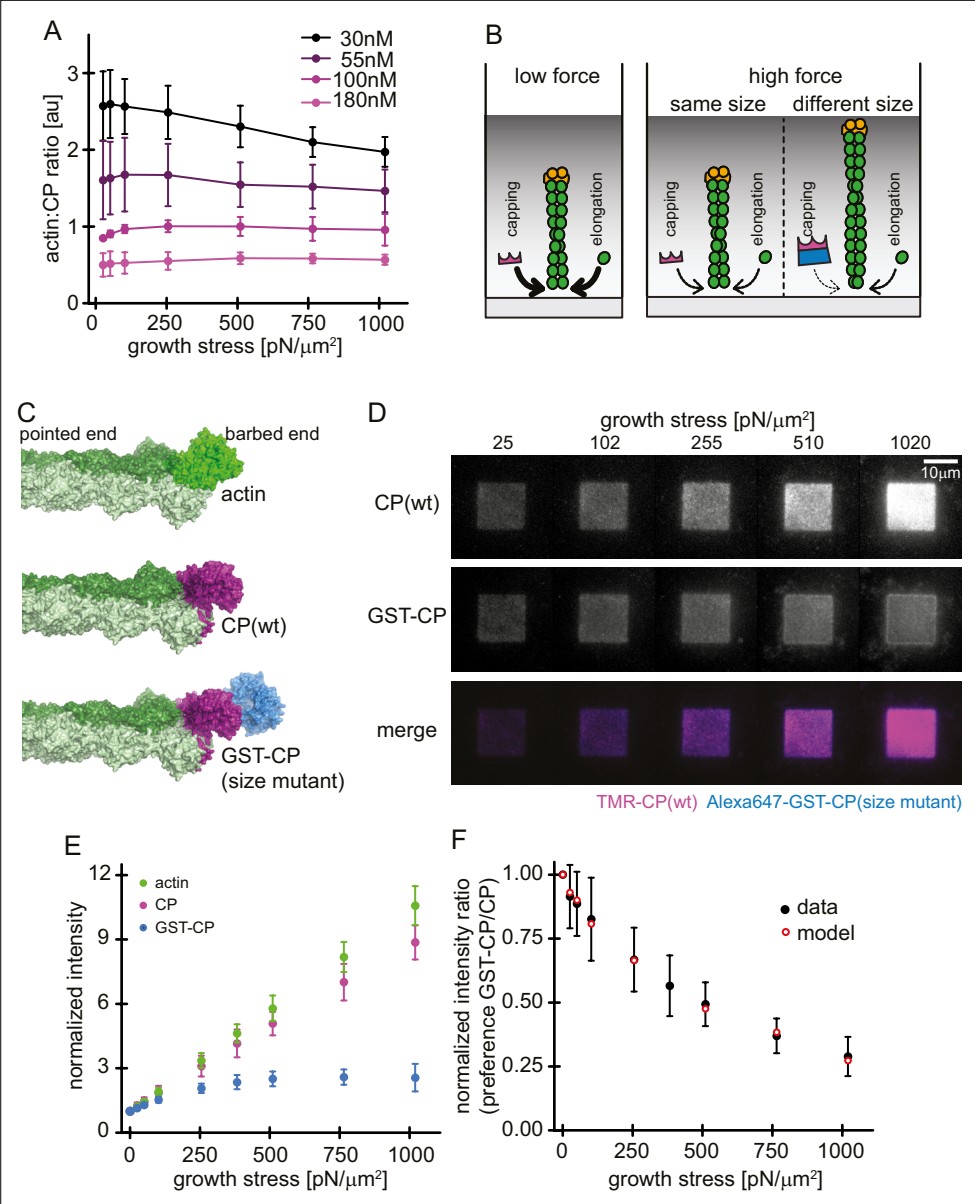

**Figure 5.** Load dependence of capping and a direct test of the elastic Brownian theory of force generation by actin networks. (**A**) Average ratios of capping protein to actin fluorescence for networks grown at different CP concentrations as indicated as a function of load (see *Figure 4—figure supplement 1*). Error bars are SD. N=3 independent experiments for each CP concentration used. (**B**) Illustration of the consequences of load dependence of capping and polymerization. Low load allows for high capping and polymerization rates (left panel). A similar load dependence of these two processes maintains filament length at high load (middle panel), whereas a difference in load dependence leads to changes in filament length (right panel). (**C**) Structural models of a filament barbed end (light and dark green) bound by either an additional actin monomer (top panel, bright green), a wt CP heterodimer (middle panel, magenta) or an engineered GST and CP dimer fusion ('bulky variant', bottom panel, magenta = CP, blue = GST). (**D**) TIRFM images of dendritic actin networks (top panel = TMR CP (wt), middle panel = Alexa647-GST-CP (bulky variant) and bottom panel = color merge) at indicated stress. Networks were assembled at standard conditions, except that CP (wt) concentration was 90 nM (of which 10 nM were TMR-CP) and Alexa647-GST-CP concentration was 10 nM. (**E**) Mean Alexa 488-actin, TMR-CP(wt) or Alexa 647-GST-CP (bulky variant) intensity normalized to the intensity of an adjacent unloaded network as a function of load. Error bars are SD. N=4 independent experiments. (**F**) Measured mean fluorescence intensity ratios of CP(wt)/ GST-CP(bulky variant) normalized to the intensity ratio of an adjacent unloaded network as a function of load. Error

*Figure 5 continued on next page*

*Figure 5 continued*

bars are SD. N=4 independent experiments. Red open circles are derived from the Brownian Ratchet Model (see Appendix 1).

The online version of this article includes the following source data and figure supplement(s) for figure 5:

**Source data 1.** Quantification of concentration and load dependence of filament capping.

**Figure supplement 1.** Characterization of network assembly at various CP concentrations.

**Figure supplement 1—source data 1.** Quantification of network assembly at various CP concentrations.

**Figure supplement 2.** Characterization of CP variants.

**Figure supplement 2—source data 1.** Quantification of capping by bulky variants compared to wildtype capping protein.

---

protein-protein interaction (*Weisel et al., 2003*). These data demonstrate that the addition of capping protein to the free barbed end of actin filaments in a branched network is a size-dependent insertional process, whose force-dependent kinetics are described by Brownian Ratchet models (*Mogilner and Oster, 2003*; *Mogilner and Oster, 1996*).

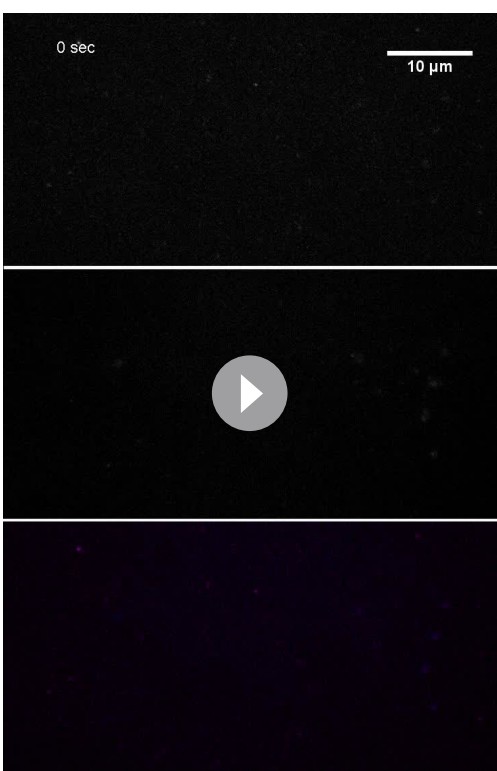

**Video 3.** Load dependence of capping in growing branched networks as visualized by differential incorporation of wildtype or bulky capping protein. Time-lapse movie from TIRF microscopy (top panel = TMR CP (wt), middle panel = Alexa647-GST-CP (bulky variant) and bottom panel = color merge with TMR-CP in magenta and Alexa647-GST-CP in blue) of dendritic actin networks either subjected to a stepwise increase in load (left square) or an adjacent unloaded control network (right square). Networks were assembled at standard conditions, except that CP (wt) concentration was 90 nM (of which 10 nM were TMR-CP) and Alexa647-GST-CP concentration was 10 nM. Scale bar = 10 μm. Conditions are as in Figure 5D.
https://elifesciences.org/articles/73145/figures#video3

## Discussion
### How force increases the number of free barbed ends

We have demonstrated that compressive forces increase the density of growing filament ends at the leading edge of a branched actin network by decreasing the (per-filament) rate of capping, and *not* by increasing the overall rate of nucleation. The binding of capping protein to the free barbed end of an actin filament is a bi-molecular inter-action, whose rate depends on the concentration of capping protein and the rate constant for fila-ment capping, that is $k_{cap}$[CP]. All our results are consistent with the idea that force decreases the capping rate constant, $k_{cap}$, without changing the local concentration of capping protein. Several lines of evidence argue against a change in the local concentration of capping protein at the leading edge of the network, including: (1) we do not observe a spatial gradient of incorpora-tion from the outer surface to the center of the network for any component, including wildtype and bulky capping proteins; and (2) the mesh size of even the densest actin networks generated in our experiments is too large to significantly impede diffusion of the soluble components used in our experiments (even the seven-subunit Arp2/3 complex). The latter point is demonstrated by a simple calculation. The highest density actin networks generated in our experiments contain an equivalent local concentration of ~1250 μM polymeric actin (*Bieling et al., 2016*). We calcu-late a mesh size ($\zeta$) of 45 nm for these networks from the following equation (*Schmidt et al., 1989*):

$$\zeta = \frac{1.47}{\sqrt{C_A}}$$

where $C_A$ is the actin concentration in µM, and $\zeta$ is mesh size in µm. For comparison, the Stokes radii of the largest network components are approximately ten times smaller. Wildtype and bulky capping proteins are at 3.8 nm (*Cooper et al., 1984*) and ~5 nm respectively, while the Arp2/3 complex has a Stokes radius of 5.3 nm (*Mullins et al., 1997*). These values are well below the threshold of 18 nm (0.4x $\zeta$ ; *Wong et al., 2004*), where interaction with the network begins to strongly affect diffusion.

## Does load adaptation happen the same way in all branched actin networks?

We studied load adaptation in actin networks created by WAVE1, but most cells express multiple nucleation promoting factors that activate the Arp2/3 complex in different locations and in response to different intracellular signals to create actin networks with distinct biological functions. For example, mammalian cells express multiple isoforms of WAVE as well as WASP, N-WASP, WASH, WHAMM, and JMY (*Campellone and Welch, 2010*). These proteins initiate assembly of actin networks required for pseudopod formation, cell adhesion, intracellular vesicle movement, endocytosis, phagocytosis, and autophagy. The mechanism of load adaptation that we describe here results solely from the force sensitivity of filament capping, which depends on the size of capping protein and the proximity of barbed ends to the membrane surface. Because it is independent of the details of the nucleation reaction, this mechanism should produce similar effects on filament density in all branched actin networks, regardless of which nucleation promoting factor directs their assembly. In other words, we expect that the denominator of *Equation 1* (per-filament capping rate) will, in all cases, decrease exponentially with force. The question then becomes: will the numerator of *Equation 1* (rate of nucleation per unit area) always behave the same way under load? We note that all of the 'Type I' (*Welch and Mullins, 2002*) nucleation promoting factors mentioned above share a common set of motifs that enable them to create force-generating actin networks. These include: (1) a central and acidic (CA) region that interacts with the Arp2/3 complex; (2) one or more WH2 domains that bind actin monomers and free barbed ends; and (3) a proline-rich region that binds multiple profilin-actin complexes. Previous work has demonstrated that the CA domains of different nucleation promoting factors stimulate different filament nucleation rates (*Zalevsky et al., 2001*; *Marchand et al., 2001*) and that variations in the proline-rich region confer different levels of polymerase activity (*Bieling et al., 2018*), but these kinetic differences are not likely to alter the basic force response of the nucleation reaction. In contrast, the WH2 domain can compete with capping protein to occupy free barbed ends (*Co et al., 2007*), and this appears to be the main interaction tethering the actin network to the membrane and producing significant 'friction' that opposes movement (*Kuo and McGrath, 2000*). Because the WH2 domain is significantly smaller than capping protein, its interaction with barbed ends will not follow the same force dependence. Therefore, nucleation promoting factors that contain two (N-WASP) or three (JMY) WH2 domains might exhibit different force dependences than those that contain only one (WAVE1-3, WASP, and WASH).

## Direct test of Brownian Ratchet theory

The effect of force on $k_{cap}$ can be neatly explained by Brownian Ratchet theories developed to describe the effects of force on actin filament elongation (*Mogilner and Oster, 1996*; *Peskin et al., 1993*; *Theriot, 2000*). Since both capping protein and monomeric actin require the same sized gap between the filament barbed end and the surface it pushes against, Brownian Ratchet theory predicts that capping and elongation will respond to force in the same way. This means that, when the rate of filament elongation slows under load, the rate of capping also slows by the same amount. Filaments grow slower but they grow for a proportionally longer time and reach the same average length (*Bieling et al., 2016*). We directly tested a central prediction of the Brownian Ratchet model by creating a 'bulky' mutant capping protein that requires a larger sized gap to bind the end of an actin filament. When we applied force to networks that contained the mutant capping protein, the rates of filament elongation and capping diverged sharply. This experiment illustrates the functional importance of the capping protein force response and provides the most direct experimental support for Brownian Ratchet models of force generation to date.

The matched force responses of filament elongation and capping suggest that the size of capping protein may be evolutionarily adaptive and subject to positive selection. A mismatch between the sizes of capping protein and actin would cause filaments to lengthen or shorten in response to force, both of which could be deleterious. Filament lengthening would impair force generation because longer filaments more easily bend and/or buckle, while filament shortening would weaken the network by decreasing filament density and network connectivity. In this way, the force-generating Brownian ratchet (filament elongation) is controlled by a second, regulatory Brownian ratchet (filament capping) to form a balanced, self-assembly motor. Load-invariance of average filament length also means that this critical network parameter can be tuned independently by polymerases such as formins or Ena/VASP proteins or even WASP-family nucleation promoting factors.

### How free barbed ends interfere with activation of the Arp2/3 complex

One unexpected result of our study is that force actually *decreases* Arp2/3 complex activity, suggesting that mother filament availability is not a limiting resource for the nucleation reaction. On the contrary, the increased number of free barbed ends generated under load inhibits nucleation by tying up WH2 domains and decreasing the ability of NPFs to activate the Arp2/3 complex. We observed this negative feedback in a previous study (*Akin and Mullins, 2008*) and attributed it to a mechanism we called 'monomer gating'. Both 'barbed end interference' and 'monomer gating' are functionally equivalent in accounting for the fact that the WH2 domain acts as the primary negative feedback regulator dialing down the filament nucleation rate in response to increased numbers of free barbed ends. The key difference is that 'monomer gating' is based on the ability of growing barbed ends to remove monomeric actin from NPFs (*Bieling et al., 2018*; *Mullins et al., 2018*), while 'barbed end interference' reflects a direct, inhibitory interaction between NPFs and the barbed ends of actin filaments (*Co et al., 2007*; *Funk et al., 2021*). Both mechanisms describe the key features of feedback regulation: (1) how the products of the nucleation reaction directly inhibit branching; (2) why capping protein stimulates Arp2/3-dependent nucleation; and (3) how branched actin networks achieve a constant, homeostatic density despite the autocatalytic nature of the nucleation reaction (*Mullins et al., 2018*). Our current study, however, provides direct experimental evidence for 'barbed end interference,' and suggests that this mechanism may account for the majority of feedback regulation in branched actin networks.

### The high failure rate of newly formed branches

By comparing Arp2/3 incorporation rates measured in bulk fluorescence versus single-molecule assays we discovered that some new branches fail soon after they are created. We know that these branch failures are limited to a region that is a few tens of nanometers from the site where the branches are created because, outside of this region the Arp2/3 complex remains stably associated with the network. These early branch failures appear to increase with applied force but, due to the nature of the TIRF illumination used in our assays, we cannot say whether the failure rate increases with force. It is formally possible that early branch failures are force-independent and that the force-induced slowing of network growth causes a larger number of these events to occur within the TIRF illumination field. Regardless, our experiments reveal that newly formed branches are surprisingly vulnerable, especially given the reported age-dependent sensitivity of branches to flow-induced shearing (*Pandit et al., 2020*). This period of vulnerability could be related to the rate of ATP hydrolysis or to the nature of forces acting on uncapped filaments that are actively growing against the coverslip surface.

## Materials and methods

### Protein biochemistry

#### Coverslip-immobilized proteins (NPFs and mCherry variants)

The coding sequence of Human WAVE1 lacking the N-terminal SH1 domain (AA 171–559, WAVE1ΔN) was codon-optimized for expression in *E. coli* (GeneArt) and fused to an N-terminal mCherry-tag harboring an N-terminal Lys-Cys-Lys-(KCK-)tag (for surface immobilization) followed by a His$_{10}$-tag (for purification) and cloned into a modified pET vector containing a TEV-cleavable z-tag (REF). To prevent surface attachment via protein sites other than the N-terminal KCK-tag, endogenous Cysteine residues of WAVE1 (Cys 296 and 407) were replaced with Serine without affecting protein activity. A

non-fluorescent version of this mCherry-NPF fusion construct was generated by introducing a Tyr71->Ser mutation in mCherry (darkCherry) to facilitate multicolor TIRF microscopy when direct visualization of the NPF was not necessary. For the detection of WH2 occupancy by FRET, a residue directly upstream of the $WH_2$ domain (Thr490 in WAVE1ΔN) was mutated to Cys and the N-terminal KCK- was substituted for a Sortase-$(Gly_5-)$tag. For the purification of mCherry or dark mCherry lacking NPF activity, we introduced a STOP codon between the Cherry and NPF moiety. Proteins were expressed in *E. coli* (Star pRARE) for 16 hr at 18 °C and purified by IMAC over a HiTrap Chelatin column, followed overnight TEV cleavage on ice, ion-exchange chromatography over a Source Q (XK 16–20) column and gelfiltration over a HiLoad Superdex 200 column. The FRET NPF construct was labeled after ion-exchange chromatography with Alexa488-Maleimide at position 490 and then subjected to sortase-mediated peptide ligation of a Cys-containing peptide (CLPTEGG) to the N-terminal Sortase-$(Gly_5-)$ tag, followed by gelfiltration for the removal of free peptide. Proteins were SNAP-frozen in liquid nitrogen in storage buffer (20 mM HEPES (pH = 7.5), 150 mM NaCl, 0.5 mM TCEP, 0.1 mM EDTA, 20% Glycerol).

## Actin

Native, cytoplasmic actin from Amoeba castellanii was purified by ion-exchange chromatography and a cycle of polymerization-depolymerization as described previously (*Hansen et al., 2013*) and stored in filamentous form dialyzing against polymerization buffer (20 mM Imidazole (pH = 7.0), 50 mM KCl, 1.5 mM $MgCl_2$, 1 mM EGTA, 0.5 mM ATP, 0.5 mM TCEP). 5 ml fractions of the filamentous pool were depolymerized at a time by dialyzing into G-Buffer (2 mM Tris-Cl (pH = 8.0), 0.1 mM $CaCl_2$, 0.2 mM ATP, 0.5 mM TCEP) for 1 week, followed by gelfiltration over a HiLoad Superdex 200 (XK16-60) column. Actin was kept in monomeric form after gelfiltration at 4 C for up to two months. Actin was fluorescently labeled with Alexa488-Maleimide at Cys 374 as previously described (*Hansen et al., 2013*). For labeling actin with Atto540Q-NHS, the profilin-actin complex was formed in G-Buffer with a 1.5-fold excess of profilin. The complex was isolated by gelfiltration over a Superdex 75 column in labeling buffer (2 mM HEPES (pH = 8.0), 0.1 mM $CaCl_2$, 0.2 mM ATP, 0.5 mM TCEP), concentrated and labeled at reactive lysine residues by incubating with a 10-fold excess of the NHS-dye for 1 hr on ice. After quenching with Tris-Cl (2 mM, pH = 8.0), actin was polymerized by addition of 10 x polymerization buffer and a small quantity (1% of total actin) of freshly sheared filaments. After polymerization for 1 hr at room temperature, filaments were pelleted by ultracentrifugation (20 min at 80 krpm in a TLA100.2 rotor) and then depolymerized in G-Buffer for 1 week in the dark. Depolymerized, labeled actin was then gelfiltered over a Superdex 75 column and stored on ice.

## Arp2/3 complex

The native, bovine Arp2/3 complex was purified from calf thymus glands (PelFreez) by a series of ammonium sulfate precipitation and ion-exchange chromatography (DEAE, Source Q and Source S) steps followed by gelfiltration (Superdex 200) as described previously (*Doolittle et al., 2013*). Arp2/3 was fluorescently labeled by addition of 3-fold excess of maleide-dye conjugate, incubated for 1.5 hr on ice, quenched by adding DTT to 1 mM and desalted into VCA Buffer A (5 mM Tris-Cl (pH = 8.0), 5 mM NaCl, 1 mM DTT, 0.2 mM $MgCl_2$, 0.1 mM ATP). To remove a small subfraction of Arp2/3, which irreversibly bound to the NPF after labeling, the complex was bound to a 5 ml NPF affinity column (N-WASP VCA immobilized on a HiTrapNHS resin) and eluted by a 10CV gradient to VCA Buffer B (5 mM Tris-Cl (pH = 8.0), 5 mM NaCl, 0.2 mM $MgCl_2$, 0.1 mM TCEP, 0.1 mM ATP). Peak fractions were pooled, concentrated and gelfiltered over a Superose 6 column. Proteins were SNAP-frozen in liquid nitrogen in storage buffer (5 mM HEPES (pH = 7.5), 50 mM NaCl, 0.5 mM $MgCl_2$, 0.5 mM TCEP, 0.5 mM EGTA, 0.1 mM ATP, 20% Glycerol).

## Capping protein

To generate wt CP, the α1 and β2 isoforms of murine heterodimeric capping protein were cloned into pETM20 and pETM33, respectively. To generate fluorescently tagged, wt CP, an N-terminal SNAP-tag (*Keppler et al., 2003*) was fused to the beta subunit. To construct larger sized CP dimers ('bulky mutants') we fused either a GST-tag (aa 1–217 of Glutathione S-transferase from Schistosoma japonicum) to the N-termini of both CP subunits or a NusA-tag (full length from *Escherichia Coli*) to the alpha subunit. GST and CP domains were separated by a $(Gly)_5$Thr- (21.5 Å) spacer, whereas

NusA and CP domains were separated by a GlyThr-(7.2 Å) spacer. GST-CPα was cloned into pETM20, whereas GST-CPβ and NusA-CPα were cloned into pETM11. To generate fluorescently tagged GST-CP, an N-terminal SNAP-tag was additionally fused to the GST- CPβ chimera. To generate fluorescently tagged NusA-CP, an N-terminal SNAP-tag was additionally fused to the NusA-CPα chimera. Proteins were co-expressed in corresponding pairs of alpha and beta subunit combinations in *E. coli* (Rosetta) for 16 hr at 18°C and purified by IMAC over a 5 ml HiTrap Chelating column followed by overnight TEV/Prescission cleavage of the N-terminal His-tags on ice. After desalting over a HiLoad Desalting column, uncleaved protein and free tags were removed by recirculation over the IMAC column. The flow through was subjected to ion-exchange chromatography over a Mono Q column and gelfiltration over a Superose 6 column. Proteins were SNAP-frozen in liquid nitrogen in storage buffer (10 mM Tris-Cl (pH = 7.5), 50 mM NaCl, 0.5 mM TCEP, 20% Glycerol). Addition of the N-terminal tags (SNAP-, GST-, NusA- or combination of those) did not affect capping activity in the absence of force as measured by polymerization of pyrene-actin in bulk (*Figure 4—figure supplement 1*) or capping in single filament TIRFM assays.

### Profilin
Human profilin 1 was expressed and purified as previously described (*Bieling et al., 2016*) and SNAP-frozen in liquid nitrogen in storage buffer (10 mM Tris (pH = 8.0), 50 mM KCl, 1 mM EDTA, 0.5 mM TCEP, 20% Glycerol).

### Ezrin-ABD
The C-terminus of human Ezrin (aa 553–586) followed by a 13aa Gly-rich linker and a C-terminal KCK-motif was cloned into pGEX-6P-2, expressed in *E. coli* (Rosetta) for 8 hr at 25°C, purified over a GST Trap column followed by overnight GST-Prescission cleavage on ice and desalting. Desalted protein was filtered over a GST Trap column to remove free GST and GST-Prescission. The flow through was gelfiltered over a Superdex 75 column and SNAP frozen in liquid nitrogen in in storage buffer (10 mM Tris (pH = 8.0), 150 mM KCl, 0.5 mM TCEP, 20% Glycerol).

### Myotrophin/VI
Full length, human myotrophin was cloned into a modified pETM11 vector containing a TEV-cleavable, N-terminal $His_{10}$-tag and expressed in expressed in *E. coli* (Rosetta) for 8 hr at 25°C, purified over a HiTrap Chelatin column followed by overnight TEV cleavage on ice and desalting. Desalted protein was filtered over a HiTrap Chelatin column to remove free His-tag and TEV. The flow through was gelfiltered over a Superdex 200 column and SNAP frozen in liquid nitrogen in in storage buffer (20 mM HEPES (pH = 7.5), 150 mM KCl, 0.5 mM TCEP, 20% Glycerol).

## Surface functionalization and protein immobilization
### Coverslip functionalization, photolithography, and protein immobilization
Glass coverslips (22 × 22 mm, #1.5, high precision, Zeiss) were functionalized and patterned as described previously (*Bieling et al., 2016*). Briefly, surfaces were rigorously cleaned by consecutive incubation in 3 M NaOH and Piranha solution (3:2 concentrated sulfuric acid to 30% hydrogen peroxide) followed by silanization with (3-Glycidyloxypropyl)trimethoxysilane. Silanized surfaces were then passivated by reacting with diamino-PEG. Subsequently, exposed amino groups were reacted with a heterobifunctional crosslinker (BMPS) to create PEG-maleimide coated coverslips, which were subjected to UV-microlithography using a chrome-on-quartz photomasks, which selectively protected maleimide groups within chrome-covered areas from UV exposure. Micropatterned PEG-maleimde coverslips were then loosely attached to flow chambers constructed of PLL-PEG passivated microscopy counter slides and thin PDMS stripes (flow cell volume = 40 µl). For the immobilization of NPF on micropatterned PEG-maleimide coverslips, protein aliquots of KCK-Cherry- WAVE1ΔN (NPF) and KCK-Cherry (mock protein) were rapidly thawed and und pre-reduced with 1 mM beta-mercaptoethanol for 30 min on ice and then desalted twice into Immobilization Buffer (20 mM HEPES (pH = 7.5), 300 mM NaCl, 0.5 mM EDTA). Protein concentration was determined by $Au_{280nm}$ and NPF protein mix was prepared by diluting desalted proteins to 10 mM total in immobilization buffer, followed by direct incubation for 25 min at room temperature with the freshly patterned PEG-maleimide coverslip in the flow cell contained in a humidified chamber. The NPF density was controlled by adjusting the relative

percentage of KCK-darkCherry-WAVE1ΔN (NPF) and KCK- darkCherry (mock protein). Coverslips were prepared using a percentage of 60% NPF and 40% mock protein. For the FRET experiments determining the WH2 occupancy, coverslips were prepared using 50% KCK-darkCherry-WAVE1ΔN, 10% CLPTE-darkCherry-WAVE1ΔN (Alexa488-Cys490) and 40% KCK-darkCherry. After protein immobilization, flow cells were washed with 6 flow cell volumes wash buffer (20 mM HEPES (pH = 7.5), 300 mM NaCl, 0.5 mM EDTA, 5 mM beta-mercaptoethanol), incubated for 3 min to quench residual maleimide groups, washed with 6 flow cell volumes storage buffer (20 mM HEPES (pH = 7.5), 300 mM NaCl, 0.5 mM EDTA, 2 mM TCEP) and stored at 4 °C in a humid container for up to 5 days.

### Cantilever functionalization

Ezrin-coated AFM cantilevers were prepared as described previously (*Bieling et al., 2016*). Tipless, uncoated cantilevers were chemically cleaned by incubating in Piranha solution (3:2 concentrated sulfuric acid to 30% hydrogen peroxide), washed, transferred to custom-built PDMS incubation chambers and functionalized by incubating for 1.5 hr in Silane-PEG5000-Maleide (Nanocs, freshly resuspended to 2% (w/w) in 95% ethanol, 5% water, pH = 5.0) at room temperature. The cantilevers were then washed twice in excess ethanol, dried for 1 hr at 75 °C and washed with ultrapure water. Ezrin-ABD was diluted to 20 μM in cantilever buffer (2 mM Tris-Cl, pH = 8.0) and immobilized on PEG-Maleimide-functionalized cantilevers by overnight incubation at 4°C in custom-built PDMS incubation chambers. Immediately before the experiment, Ezrin-coated cantilevers were washed in excess cantilever buffers and dried.

## Fluorescence and atomic force microscopy system

### TIRFM-AFM system

Imaging was performed on an Observer.Z1 (Zeiss) microscope equipped with a total internal reflection fluorescence (TIRF) slider (Zeiss), a TIRF objective (PlanApochromat 100 × 1.46 TIRFM, Zeiss) and a cooled charge-coupled device camera (iXon888, Andor). Fluorescence excitation was accomplished by three diode-pumped solid-state laser lines (488, 561, and 644 nm), which were controlled using an acousto-optical filter and coupled into a single fiberoptic light guide (custom laser launch, Spectral Applied Research). Micro-Manager (*Edelstein et al., 2010*) was used to control the shutters, acousto-optical filter, dichroic mirrors and camera. Laser intensity and exposure was minimized to avoid photo-bleaching. For bulk multi-color fluorescence measurements of dendritic network component densities, images (300ms exposure time) were taken at custom intervals of increasing time (5–30 s to avoid bleaching in networks growing with reduced velocity at elevated forces). Fast, one-color imaging of single molecules was performed at an increased frame rate of 10 frames/s and a 100ms exposure time ("streaming" mode).

Force measurements were performed using commercial AFM system (BioScope Catalyst, Bruker) modified as described in detail in *Bieling et al., 2016*. Briefly we (1) replaced the original AFM photo-diode detector with a position sensitive device (PSD, Pacific Silicon Sensor, DL100-7PCBA3) to obtain a larger dynamic force range, (2) constructed a custom cantilever holder with low cantilever angle (~3°) to closely match the idealized geometry of two parallel planes and also to prevent slippage between the AFM cantilever and actin network, (3) constructed a custom sample holder that could prevent evaporation, and (4) utilized a customized setup to perform micro-rheology measurements (*Alcaraz et al., 2003*; *Mahaffy et al., 2000*). All the AFM electronic signals from PSD were pre-processed by an electronic filter (Krohn-Hite, 3362) set to dc low-pass at 30 Hz. LabView was used for signal processing, data acquisition, and piezo stage control.

## Denditic network assembly assays

### General dendritic network assembly assay

Flow cells of micropatterned, NPF-coated coverslips were washed with twice with 250 μl of ultra-pure water (Milli-Q grade) and disassembled by removal of the coverslips. Excess water was removed by a brief (5 s) spin on a spin coater. Drying did not affect NPF activity if the coverslip was not kept in air for >30 min. The coverslip was fixated on a custom-built sample holder by adhering to a thin PDMS O-ring and the whole assembly was transferred to the microscope stage. An ezrin-coated AFM cantilever was immobilized with a drop of hot paraffin wax on a custom built cantilever holder and then attached to the AFM head, which was mounted on the microscope and lowered

to close proximity to the coverslip. 100 ml assembly buffer (20 mM HEPES (pH = 7.0), 100 mM KCl, 20 mM beta-Mercaptoethanol, 1.5 mM MgCl$_2$, 1 mM EGTA, 1 mM ATP, 0.5 mg/ml beta-casein, 10 nM Alexa488-labelled actin) were added in between coverslip and cantilever holder. Low amounts of labelled actin were included in the buffer to visualize the NPF patterns indirectly via the binding of actin monomers. Eighty μl of mineral oil containing 20 mg/ml Cithrol DPHS (to passivate the oil-buffer interface) was overlaid onto the buffer to seal if from air exposure. The Optical Lever Sensitivity (OLS) is characterized by measuring the force-distance curve in contact with the hard glass surface, prior to every measurement. An NPF pattern was then positioned at an axial distance of 3 μm directly under the AFM cantilever via the motorized stage and the x- and y-piezoelectric stage control. Actin network growth was finally initiated by addition of 50 μl of network proteins in assembly buffer (final concentration: 5 μM actin, 5 μM profilin, 100 nM Arp2/3, 100 nM CP if not indicated otherwise). Synchronously, multicolor TIRFM time-lapse imaging was initiated. For bulk fluorescence, multicolor TIRFM experiments, the protein mix was supplemented with 1%Alexa488-actin, 5% Alexa647-Arp2/3% and 15% TMR-SNAP-CP. After the height of the growing network reached the cantilever (as indicated by cantilever displacement and a rise in force), the force was kept constant at a defined setpoint by engaging the force-feedback mechanism ('force-clamp mode'). The force was maintained until both network fluorescence and growth velocity reached a steady state, upon which the force was changed to a higher setpoint. This cycle was repeated until network growth was slowed to velocities <200 nm/min, close to mechanical stall. Network growth did not exhibit hysteresis, hence the order or duration by which the individual forces were applied did not affect the growth velocity.

## Single molecule dendritic network assembly assay for TIRFM-AFM

Assays were carried out as described in the section 4.1 with the following exceptions: For single color, single molecule imaging, assembly buffer was supplemented with an oxygen scavenger system (40 mM glucose, 125 μg/ml glucose oxidase, 40 mg/ml catalase) and 2 mM Trolox and the protein mix contained 0.02% (1 in 5000) Alexa647-Arp2/3.

## FRET assay for the determination of the WH2 occupancy of the NPF

Assays were carried out described in section 4.1 with the following exceptions: Coverslips were functionalized using a low percentage of the NPF FRET construct (CLPTE-darkCherry-WAVE1 ΔN(Alexa488-Cys490)), see section 2.1. Reactions were scaled down to 100 μl volume (in comparison to 150 μl for standard conditions). Assembly buffer was supplemented with an oxygen scavenger system (40 mM glucose, 125 ug/ml glucose oxidase, 40 mg/ml catalase), 2 mM Trolox and the protein mix contained only Alexa647-Actin (1% of total) as fluorescent label. The load was maintained at 1020 Pa until the network reached steady state growth and a minimum height of >3 μm. Network growth was then arrested and capping was inhibited by carefully diluting the reaction (100 μl total) by adding 200 μl fixing buffer assembly buffer containing Latrunculin B (15 μM final), Phalloidin (15 μM final), Myotrophin (5 μM final, competitive, high-affinity CP inhibitor *Bhattacharya et al., 2006*), profilin (7.5 μM final) and Atto540Q-actin (7.5 μM final). The fraction of quencher-labeled actin (Atto540Q-actin) after arrest was thus 7.5 μM of 10 μM total. For control experiments in the absence of a dendritic network, the assembly buffer was supplemented with 15 μM Latrunculin B before the experiment to prevent actin polymerization.

## 'Spike-in' experiments using larger sized capping protein variants ('bulky mutants')

Experiments were carried out as described in section 4.1 with the following exceptions: The overall CP pool (100 nM) consisted of 90% unlabeled, wt CP, 5% TMR-SNAP-CP and either 5% Alexa 647-SNAP-GST-CP or 5% Alexa 647-SNAP-NusA-CP. This limited the influence of the lowered capping rate of the size mutant at elevated forces on the overall network assembly kinetics.

## Data analysis
### Quantification of network growth velocity
Constant growth forces were applied to a growing network under AFM force clamp control. The growth velocity of the network was determined by the slope of height-time curve at individual constant growth forces. However after switching to a new growth force, the network needed time to adapt the

new growth force to reach constant growth. Therefore, the slope is not considered for growth velocity until the network reached the steady constant growth where the slope is constant.

## Quantification of bulk fluorescence intensities from TIRFM and confocal imaging

The mean intensities of all network components (actin, Arp2/3, CP) from multicolor, time-lapse TIRFM images were quantified via ImageJ (ROI Manager->Multi Measure function) from square region of interests (ROIs) matching the network area. Background intensity was determined from adjacent regions (10 µm distance) of the same size and subtracted from the network intensity. For Arp2/3 and actin, a small (<30% of total intensity in the absence of force for TIRFM imaging,<5% for confocal imaging) fraction of fluorescence in the network area is due to binding to the NPF in addition to the actin network. The intensity of this signal was quantified during the initial lag phase preceding actin network nucleation and subtracted from the network intensity. The fluorescence intensities were plotted as a function of time together with the height of the sample as well the counterforce. Mean fluorescence intensities at were then calculated by averaging over the fluorescence signal during steady growth at a constant force. The variance in fluorescence intensity during these steady state periods was very low (SD <2% total).

## Single molecule tracking and classification

For detection and tracking of single Arp2/3 molecules in TIRF time-lapse images, we used the u-track software package (*Jaqaman et al., 2008*). After complete tracking, an additional step classifies tracks into productive (molecules that are incorporated into the network and continuously grow out of the TIRF microscopy field of view as indicated by a progressive drop in fluorescence intensity) or unproductive (stuck and/or blinking molecules at constant intensity and position). This is done in a semi-automated process: All individual tracks of minimum length 5 frames (=500ms) are randomly distributed amongst six biological experts. Each expert subsequently classifies all individual tracks of his share as productive or unproductive. In this step, 10% of all tracks are classified by two experts independently to estimate the classification uncertainty. This led to a fraction of tracks of at least 90% over all sample clips that is associated to the same class by both experts. In order to efficiently process significant amounts of microscopy data, we further automated our analysis procedure, by calculating a set of 10 feature parameters for each track. All features are based on the dynamics of intensity and position of each track. The set of features in combination with the combined classification results of the experts was used subsequently for training a supervised random forest classifier. Cross validation yielded correct classification in at least 82% of all tracks. In order to further improve this performance, we followed an active learning strategy in which borderline cases (i.e. tracks for which the decision trees in the forest do not agree well in their classification decision) are decided by an expert. Using cross validation of this semi-automated procedure, the amount of manually classified tracks is tuned to yield a comparably high correct classification performance as the group of experts (i.e. >~90%). After classification, productive tracks are used in further analyses exclusively. MATLAB code for this analysis is available on Github (*Weichsel, 2022*; copy archived at swh:1:rev:5615adf7504954ed42f47e2d2fa37465a19bf6c6).

## Determination of bulk nucleation rates from single-molecule calibration experiments

The mean event rates of productive network incorporation (in counts per network per second) was determined for Arp2/3 from single molecule 'spike-in' experiments at 25 Pa and multiplied by the respective labeling ratio to yield the total nucleation rate at this load.

## Quantification of the WH2 occupancy of the NPF by FRET TIRFM imaging

The fluorescence of donor-labeled NPF (CLPTE-darkCherry-WAVE1ΔN(Alexa488-Cys490)) was plotted as function of time after addition of quencher-labeled actin (Atto540Q-actin). The data was fitted to a sum of two exponential functions: with the rapid phase caused by FRET (binding of quencher-labeled actin to the WH2 domain of the NPF) and the slow phase attributed to bleaching of the donor. The bleaching rate ($k_2$) was independently determined in control experiments and

fixed when fitting the FRET data. The amplitude of the fast, FRET phase ($I_{FRET}$) was determined for three cases: (a) In the absence of a dendritic network, (b) in the presence of a dendritic actin network that was assembled in the absence of force (0 Pa) or (c) in the presence on a dendritic actin network that was assembled at a defined load force of 1020 Pa. Assuming that all WH2 domains are free to interact with monomers in the absence of a dendritic network (a), we calculated the amount of blocked, occupied WH2 domains in the other cases (b and c) by the relative decrease in $I_{FRET}$.

## Acknowledgements

We thank Scott Hansen for reagents, discussions, and comments on the manuscript, Minhajuddin Sirajuddin for help with creating the structural model of bulky CP, and members of the DAF and RDM labs for discussions. This work was supported by NIH R01 GM134137 (DAF), NIH 1R35 GM118119 (RDM), HHMI (RDM), HFSP LT-000843/2010 (PB), HSFP CDA00070/2017-2 (PB) and EMBO ALTF 854–2009 (PB). T-DL was supported by the Taiwan National Science Council. DAF is a Chan Zuckerberg Biohub Investigator.

## Additional information

### Funding

| Funder | Grant reference number | Author |
|---|---|---|
| National Institutes of Health | 1R35 GM118119 | R Dyche Mullins |
| National Institutes of Health | R01 GM134137 | Daniel A Fletcher |
| Howard Hughes Medical Institute | | R Dyche Mullins |
| Human Frontier Science Program | LT-000843/2010 | Peter Bieling |
| Human Frontier Science Program | CDA00070/2017-2 | Peter Bieling |
| European Molecular Biology Organization | ALTF 854-2009 | Peter Bieling |
| Chan Zuckerberg Initiative | | Daniel A Fletcher |

The funders had no role in study design, data collection and interpretation, or the decision to submit the work for publication.

### Author contributions

Tai-De Li, Conceptualization, Data curation, Formal analysis, Methodology, Writing - original draft, Writing - review and editing; Peter Bieling, Conceptualization, Data curation, Formal analysis, Investigation, Methodology, Writing - original draft, Writing - review and editing; Julian Weichsel, Formal analysis, Software; R Dyche Mullins, Daniel A Fletcher, Conceptualization, Data curation, Formal analysis, Funding acquisition, Investigation, Project administration, Resources, Supervision, Writing - original draft, Writing - review and editing

### Author ORCIDs

Peter Bieling http://orcid.org/0000-0002-7458-4358
R Dyche Mullins http://orcid.org/0000-0002-0871-5479
Daniel A Fletcher http://orcid.org/0000-0002-1890-5364

### Decision letter and Author response

Decision letter https://doi.org/10.7554/eLife.73145.sa1
Author response https://doi.org/10.7554/eLife.73145.sa2

# Additional files

## Supplementary files

• Transparent reporting form

## Data availability

Source data files have been provided for the top and bottom panels of Figure 1c. Videos 1–3 contain source data for Figures 1–4.

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

## Appendix 1

### Brownian Ratchet Model of branched network assembly

Based on the original Brownian Ratchet model for actin assembly (*Peskin et al., 1993*), the actin monomer association rate to filament free barbed ends growing against a load can be written a function of applied force as

$$R = \delta \left( \alpha e^{-\frac{f \cdot \delta}{K_B T}} - \beta \right) \tag{1}$$

, where δ is the Brownian-Ratchet gap size required for the addition of an actin monomer, α is the association rate in the absence of force, *f* is the force on an individual actin filaments, $K_B$ is the Boltzmann constant, *T* is the temperature in Kelvin and *β* is the dissociation rate. We assume that the addition of capping protein to the filament end is a very similar insertional process that, like monomer addition, requires an opening of a gap. The rate of capping should, therefore follow the same equation. For two differently sized capping proteins (wt CP and GST-CP) as in our experiment, we can relate their relative rate of incorporation (R) to their characteristic gap size (δ) via

$$\frac{R_{GST}}{R_{WT}} = \frac{\delta_{WT}}{\delta_{GST}} \cdot e^{\frac{-f(\delta_{WT} - \delta_{GST})}{K_B T}} \tag{2}$$

For filaments pushing against the load in orientations different from a normal (90 °C) angle, the filament contact angle (*θ*) has to be taken into account and the normalized ratio can be written as

$$e^{\frac{-f(\delta_{WT} - \delta_{GST}) \cdot sin\theta}{K_B T}} \tag{3}$$

We can obtain the compressive force on an individual, growing filament via

$$f = \frac{F_{total}}{N_{load}^{BE}} \tag{4}$$

where $F_{total}$ is the total force generated by the number of free barbed ends sharing the load ($F_{total}$). Within a branched network, the free barbed ends associating with the WH2 domain of the NPF are neither involved in elongation nor capping and thus do not contribute to active force generation. Therefore, the average force on individual filaments has to be re-written as

$$f = \frac{F_{total}}{N_{total}^{BE} - N_{WH2}^{BE}} \tag{5}$$

where the $N_{total}^{BE}$ is the total number of free barbed ends and $N_{WH2}^{BE}$ is the number of free barbed ends associating with the WH2 domain of the NPF. Note that both of these quantities change when the network experiences external load (*Figure 3*, *Bieling et al., 2016*).

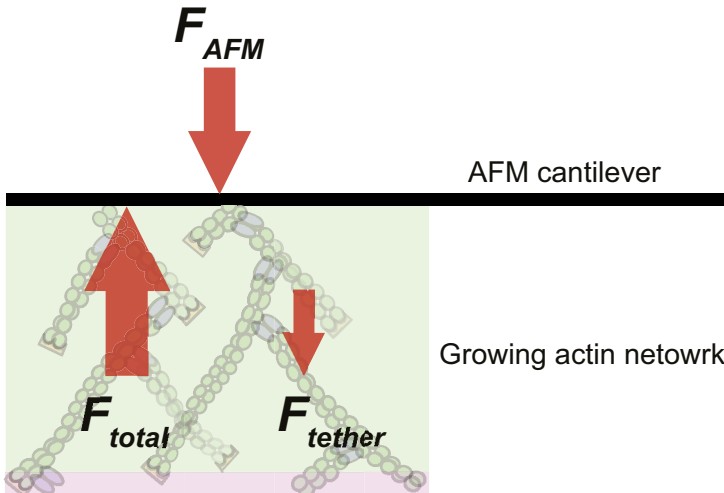

**Appendix 1—figure 1.** Scheme illustrating the balance of forces in our experimental setup. Branched network assembly produces a protrusive force ($F_{total}$), which is counteracted by (i) the external load force applied by the AFM cantilever ($F_{AFM}$) and (ii) the internal frictional forces that originate from attractive tethering forces due to interactions between the network and the NPF-coated surface ($F_{tether}$).

Finally, we assume that a filament end bound to the WH2 domain of the NPF exert a pulling or tether force ($f_{tether}$). This tethering force scales with the number of tethered ends and adds to the externally applied force via the AFM cantilever to resist movement. Therefore, the total load on the growing free barbed ends will be the sum of the total tethering force and applied AFM force:

$$F_{total} = F_{AFM} + F_{tether} = F_{AFM} + f_{tether} \cdot N_{WH2}^{BE}, \tag{6}$$

Combining *Equations 3; 5; 6*, the normalized GST-CP to wt CP rate ratio becomes

$$Normalized\,rate\,ratio \sim e^{-\frac{F_{AFM} + f_{tether} \cdot N_{WH2}^{BE}}{K_B T \cdot \left(N_{total}^{BE} - N_{WH2}^{BE}\right)} (\delta_{WT} - \delta_{GST}) \cdot sin\theta} \tag{7}$$

All of these quantities, with the exception of the characteristic tethering force can be well estimated based on our measurements here and in *Bieling et al., 2016* as detailed in the following sections.

**Total number of free barbed ends** ( $N_{total}^{BE}$)

To obtain the number of free barbed ends within the actin network ($N_{total}^{BE}$), we are using the previously measured actin polymerization rate per network area ($P_{actin}$ monomer/s/µm²) and the measured network growth velocity ($V_{network}$ um/s) according to

$$V_{network} = \frac{P_{actin} \cdot A}{N_{total}^{BE}} \cdot \delta_{eff}, \tag{8}$$

where $A$ is the network growing area and $\delta_{eff}$ is the effective ratchet size for one monomer association. For $P_{actin}$ we found a value of 7,135 monomers/s/um² and $V_{network}$ is 5.69 um/min at 25 Pa (*Bieling et al., 2016*). The average contact angle between the filament ends and the load in the absence of external load can be assumed to be 54°, as a result of the characteristic angle of Arp2/3 branching. We therefore estimate $\delta_{eff} = 2.7nm \cdot Cos\left(\frac{72^o}{2}\right)$. In our experiment setup, the network area is, 14 × 14 µm². Therefore, the total number of growing filaments (free barbed ends) of the whole network at 25 Pa can be calculated as 32,211. The number of free barbed ends as a function of growth force was then obtained by scaling the measured relative density of free ends with this value (*Figure 1C*, *Bieling et al., 2016*).

## Filament contact angle ($\theta$)

The contact angle between actin filaments and the load changes in response to force to increase the actin density within the network (*Bieling et al., 2016*; *Mueller et al., 2017*). To calculate the filament contact angle, we are going to use the concept that the $N_{filament} \cdot Cos(\theta) \propto D_{actin}$ , where $N_{filament}$

is the number of filaments and $D_{actin}$ is the actin density of the network. Also, in the growing actin network, the number of total filament is defined by the number of free barbed end ($N_{total}^{BE}$). Therefore, the contact angle ($\theta$) can be written as

$$Sin\left(\theta\right) = Sin\left(\theta_0\right) \cdot normalized\left(\frac{D_{actin}}{N_{total}^{BE}}\right) \tag{9}$$

where the $\theta_0$ is the contact at zero force (54°). Both the $D_{actin}$ and $N_{total}^{BE}$ can be derived from data (Figure 3D in *Bieling et al., 2016*).

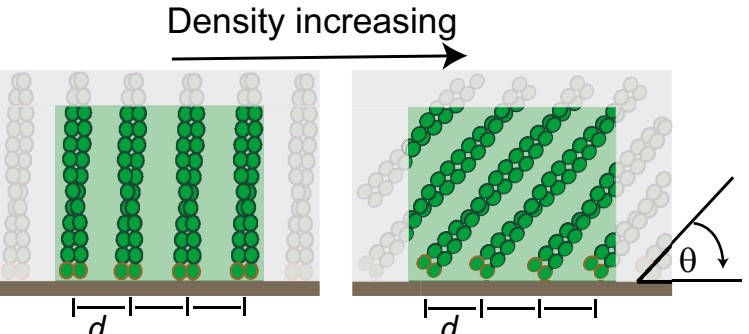

**Appendix 1—figure 2.** Illustration of the architectural changes in branched actin due to load. Filament reorientation to shallower contact angles ($\theta$) at higher loads leads to an increase in total filament density within the network.

## Number of free barbed ends associating with WH2 ( $N_{WH2}^{BE}$ )

We show that the total WH2 occupied by the free barbed ends increases from 7.2% to 26.8% when the force increases from 0 to 1020 Pa (*Figure 3*). For simplicity, we assume a linear force dependence to calculate the number of attached ends according to

$$N_{WH2}^{BE} = \left(\left(\frac{26.8\% - 7.2\%}{250 - 0}\right) \cdot F_{AFM} + 7.2\%\right) \cdot N_{total}^{WH2}, \tag{10}$$

where $N_{total}^{WH2}$ is the total available WH2 on the surface.

## Tethering force of individual free barbed end-WH2 association ( $f_{tether}$ )

The tethering force of free barbed end-WH2 association has been considered as external load-dependent in an exponential way [4]. For simplicity, we use single exponential decay in our model with

$$f_{tether} = f_{tether}^0 \cdot e^{-aF_{AFM}} \tag{11}$$

where $f_{tether}^0$ is the tethering force at zero load and $a$ is the exponential decay constant.

The ratchet gap size for wild-type CP ($\delta_{WT}$) and GST mutant CP ($\delta_{GST}$) are 2.7 nm and 7 nm according to structural models (see Materials and methods, *Funk et al., 2021*; *Kim et al., 2010*; *Narita et al., 2006*). At this point, the $N_{total}^{WH2}$, $f_{tether}^0$, and $a$ are the free parameters in *Equation 7*. The best fit of this model to our experimental data (see *Figure 4F*) yields $N_{total}^{WH2} \sim 230000$, $f_{tether}^0 \sim 0.3pN$, and $a \sim 0.025$. The first number is in a good agreement with previous data (*Bieling et al., 2018*). The tethering force has not been measured up to this point but 0.3 pN is in the range of weak protein-protein interactions; the exponential decay constant of this tethering force will need further quantitative investigation.

