## [Editor Report]

Through the use of an elegant experimental setup, this study offers a molecular explanation for why branched actin filament networks, similar to those encountered in migrating cells, become denser when growing against a mechanical load. Importantly, the results also confirm the Brownian ratchet model for actin assembly. This study captures several important features of branched filament networks and should become a reference in the field.

---

## [Decision Letter]

**Decision letter after peer review:**

Thank you for submitting your article "The molecular mechanism of load adaptation by branched actin networks" for consideration by *eLife*. Your article has been reviewed by 3 peer reviewers, one of whom is a member of our Board of Reviewing Editors, and the evaluation has been overseen by Anna Akhmanova as the Senior Editor. The reviewers have opted to remain anonymous.

Essential revisions:

1) You will see from the individual Reviews below that an effort should be made in the presentation of the results. Presented as such, even experienced Reviewers struggled sometimes to understand how the experiments were performed and analyzed. We would like you to follow their recommendations, which you will find in their respective reviews. In particular, please:

a. provide examples of the original data as they were recorded over time in the Supplementary data (Reviewer 1 Main comment 1). We agree that it is important to know the time scales at which these experiments are performed.

b. provide all explanations of how data were acquired and analyzed in this paper without referring to your Cell paper (Reviewer 2 and 3). Readers should be able to find all the necessary information here.

c. Following the discussion with the Reviewers, a suggestion could be to have a specific section where the formulas used to calculate each variable would be explicitly written. This would avoid readers to search, sometimes in the results, sometimes in the Figure Legends, or sometimes in your Cell paper, the way each of the values was calculated.

d. please show your control that network growth does not exhibit hysteresis (Reviewer 1 Main comment 5).

2) Reviewer 2 rightly points out that the rationale behind dividing by density the rate of capping and the rate of elongation, but not the rate of branching, is not explained. There are different ways to see the problem, but one could be that because all of these reactions have nearly the same substrate (capping and elongation take place at the barbed end, and branching, for geometrical reasons, takes place on the side of the filament near the barbed end), it makes equal sense to consider each rate 'per filament'. Could you please answer to this comment? It is also correct that the Y axis of Figure 5E has a unit.

In line with this comment, the suggestion of Reviewer 1 to provide single molecule data for capping protein would confirm whether the rate of capping decreases faster than nucleation under stress or not. Such data would also be nice to confirm that capping occurs only at the surface.

3) Related to point 1), it is true that experiments performed at different concentration of CP can only be compared if the pool of protein has not had time to evolve differently between the time when the proteins are mixed and the time when the measurements are made. Please reply to this comment (Reviewer 1 comment 3).

4) Regarding diffusion vs. ratchet to explain the effect of size on CP, could you provide a good argument or an easy experiment to justify that 'bulky-CP' diffuses at the same rate through branched networks as regular CP does? (Reviewer 1 Main comment 4)

All three reviewers have other interesting suggestions, those particularly deserving of your attention being:

- Please cite Wiesner et al. JCB 2003. We agree that their interpretation of comet density change under force was different than yours, but it was nevertheless the first observation of this phenomenon. This paper also nicely showed that branch density increases when capping is increased.

- Interesting comments from Reviewer 2 (second and third recommendation) and 3 (whether the conclusions are universal for all Arp2/3 complex activators) to expand the discussion. Experiments with other Arp2/3 activators and actin regulators would be great, but represent too much work for this paper.

- Comments of Reviewer 3 about data representation and analysis (justification of Student's t tests; data for Arp2/3 dwell times at 25 Pa; comment about plot shape in Figure 2D and 2F)

- Reviewer 2 and 3 both thought that Figure Supplement 1 should be part of a main Figure.

*Reviewer #1 (Recommendations for the authors):*

The study suffers from a few problems that make interpretation of the data sometimes complicated. The authors should seriously consider the comments below before publishing this study.

Main comments:

1. Experiment of Figure 1: the authors must provide more details about the experiment, and in particular in terms of kinetics. The authors need to present how the fluorescence intensity signals of actin, CP and Arp2/3 change in time over the course of an experiment. On the same graph, the displacement of the cantilever (or the growth velocity) and the growth stress over time should be shown. These original data should be presented as a Supplementary Figure for different growth stress so that readers can evaluate how experimental data were analyzed in this work. These original data should be provided from the moment when the polymerization reaction is initiated so that readers can have an idea of the time intervals between the moment when the polymerization reaction is initiated and the moments when a/ fluorescence signals of actin, Arp2/3 and CP start being visible; and b/ the cantilever starts being deflected.

2. The single molecule experiments in Figure 2 are a very nice way to check the consistency of the nucleation and capping rate measurements of Figure 1. The authors should provide single molecule data for capping protein to confirm that the rate of capping decreases faster than nucleation under stress.

3. From the moment when proteins are mixed to the moment when measurements are made, how does the protein system evolve from the initial state (which is if I understand correctly 5uM G-actin, 5uM profilin, 100nM CP and 100nM Arp2/3)? At this high G-actin concentration and even with equimolar profilin, I expect that a non-negligible fraction of actin will have polymerized within seconds/minutes, and a fraction of CP and Arp2/3 will be bound to F-actin in the bulk. Therefore, experiments performed at different CP concentration could have different temporal evolutions of bulk composition and should not be simply compared side-by-side as done here. Do the authors have measurements of free G-actin/profilin, CP and Arp2/3 concentrations at times when experiments are recorded?

4. The 'bulky' capping protein mutant experiment is exciting but another interpretation of this result could be that larger capping proteins diffuse more slowly through dense branched actin networks. The authors need to provide compelling evidence to rule out this possibility.

5. That network growth does not exhibit hysteresis is quite important. The authors should add their data proving this point as a Supplementary Figure.

*Reviewer #2 (Recommendations for the authors):*

Comparing the capping rate per filament to the nucleation rate per surface area is very misleading. I think this will confuse readers (how can the rates of nucleation and capping differ since we are at steady state?). This would really be a pity because, otherwise, this paper gives a comprehensive picture of mechano-chemical coupling in branched networks, and would be a go-to article for anyone interested in understanding this important mechanism.

So my first recommendation is to present results differently, and clearly say that the drop in nucleation and capping are the same. The current Figure 1D gives the impression that capping and elongation are affected more strongly by load, compared to nucleation, but this is only because these curves have been divided by the barbed end density, while the curve for nucleation has not. Likewise, in Figure 1E, the ratio of rates (nucleation/capping) is not unit-less, it is a density. As expected, it is very similar to the barbed end density shown in Figure 1C-bottom, determined independently. Things would be much clearer if all this was made explicit.

My second recommendation, again for clarity, is to discuss the possible differences between the mechanisms causing the drop in the rates of capping (and elongation) and branching. I understand that there is no reason to invoke a Brownian ratchet for nucleation (since Arp2/3 is bound to the NPF on the surface and does not need to fit in a gap between the filament and the surface) but this could be said explicitly, as several readers might miss this point. Also, I agree that the 'barbed end interference' mechanism seems to quantitatively account for the full drop in nucleation rate, but perhaps other differences between the capping and branching reactions could be discussed (for example: CP and actin diffuse from an infinite reservoir in solution, while active Arp2/3 is a finite surface-immobilized reservoir).

My third recommendation would be that the authors take the analysis one step further and provide, based on their results, an answer to the question they address: 'how load forces increase the number of growing filaments'. I did some calculations, and I think it can easily be done by considering the different density-dependence of the two molecular mechanisms. If the authors would rather leave this out of their paper, I would be happy to write a commentary on their article, where I would highlight their results and propose my interpretation in more detail.

*Reviewer #3 (Recommendations for the authors):*

There are places that could be clarified to better present the data or avoid confusions.

---

## [Author Response]

Essential revisions:1) You will see from the individual Reviews below that an effort should be made in the presentation of the results. Presented as such, even experienced Reviewers struggled sometimes to understand how the experiments were performed and analyzed. We would like you to follow their recommendations, which you will find in their respective reviews. In particular, please:a. provide examples of the original data as they were recorded over time in the Supplementary data (Reviewer 1 Main comment 1). We agree that it is important to know the time scales at which these experiments are performed.b. provide all explanations of how data were acquired and analyzed in this paper without referring to your Cell paper (Reviewer 2 and 3). Readers should be able to find all the necessary information here.c. Following the discussion with the Reviewers, a suggestion could be to have a specific section where the formulas used to calculate each variable would be explicitly written. This would avoid readers to search, sometimes in the results, sometimes in the Figure Legends, or sometimes in your Cell paper, the way each of the values was calculated.d. please show your control that network growth does not exhibit hysteresis (Reviewer 1 Main comment 5).

We reorganized the presentation of the results along the lines suggested by the reviewers. In response to (a) we now provide raw time course data in Figure 1C. Among other things these data demonstrate that the network returns to a new ‘steady-state’ assembly phase following each step change in force. For (b) we provide a more comprehensive description of the experimental set up in the Results section. We also provide an expanded description of data collection methods and analysis. For (c) we provide additional mathematical derivations in the Results section that should address some of the misunderstandings behind the reviewers’ second “Essential Revision” comment (below). The question about hysteresis raised in point (d) was addressed in a previous paper (Bieling et al., 2016). We reference these data (S3B from that paper) in the revised manuscript. Finally, we have moved data on the force sensitivity of filament elongation, nucleation, and capping from the supplement data into main Figure 1.

2) Reviewer 2 rightly points out that the rationale behind dividing by density the rate of capping and the rate of elongation, but not the rate of branching, is not explained. There are different ways to see the problem, but one could be that because all of these reactions have nearly the same substrate (capping and elongation take place at the barbed end, and branching, for geometrical reasons, takes place on the side of the filament near the barbed end), it makes equal sense to consider each rate 'per filament'. Could you please answer to this comment? It is also correct that the Y axis of Figure 5E has a unit.In line with this comment, the suggestion of Reviewer 1 to provide single molecule data for capping protein would confirm whether the rate of capping decreases faster than nucleation under stress or not. Such data would also be nice to confirm that capping occurs only at the surface.

The explanation of our analysis was obviously not as clear as it should have been and, as noted, we provide more detailed derivations for our mathematical methods in the revised manuscript. Regarding this specific point, Reviewer 2 rightly notes that “…for both capping and branching, the authors find that they decrease the same way with increasing loads – as they should: this is imposed by their being at steady state, where the birth rate of growing barbed ends (branching) must match their death rate (capping).” This steady state condition is actually the basis for our analysis. At steady state the overall rates of nucleation and capping must be equal (R_cap_ = R_nucleate_). Importantly, the overall rate of nucleation is a complicated function that depends on the occupancy of the WH2 domains, the surface-associated Arp2/3 complex, and the local density of polymeric actin. On the other hand, filament capping in our system appears to be a simple bimolecular interaction between soluble capping protein and free barbed ends. We demonstrated this by showing that the average filament length (i.e. the ratio of polymeric actin to capping protein in the growing network) varies as a simple inverse function of the capping protein concentration (Figure 5A of the revised manuscript). This means that the overall rate of nucleation (R_nucleate_) must equal the product of the capping protein concentration ([CP]), the surface density of free barbed ends (E), and an appropriate capping rate constant (k_cap_). This yields,

k_cap_*[CP]*E = R_nucleate_ Eq. 1

Which can be rearranged to give the density of free barbed ends,

E = R_nucleate_/(k_cap_*[CP]) Eq. 2

As the reviewer notes, the value of E is a density (namely the surface density of free barbed ends), not a unitless number. Note that a sudden decrease in per-filament capping rate (k_cap_*[CP]) with no change in overall nucleation rate will cause the number of free barbed ends to increase until the overall rate of capping (k_cap*_[CP]*E) once again matches the overall rate of nucleation. This equation is an “iron law” imposed by the steady-state (or quasi-steady state) character of the system, and it means that any increase in the density of free barbed ends must reflect EITHER an increase in the overall rate of nucleation OR a decrease in the per-filament capping rate (or possibly both). Our direct measurements of the overall nucleation rate (the quantity in the numerator) absolutely rule out the first possibility, meaning that the per-filament capping rate MUST go down with applied force. Furthermore, our measurements demonstrate that this capping rate displays the same force sensitivity as actin filament elongation. The best explanation for this phenomenon is that the *rate constant* that governs filament capping under load exhibits the same force sensitivity as the rate constant that governs addition of an actin monomer. This could have been predicted from Brownian Ratchet theory, but as the reviewer points out, it was not. Our “bulky capping protein” experiments (Figure 5) are a direct test of whether a Brownian Ratchet mechanism can account for the force sensitivity of filament capping, and they demonstrate that it can.

We trust that the above explanation clears up two of Reviewer 2’s comments: (1) “… it is worth mentioning that original models considered that force could slow filament elongation at constant nucleation and capping, therefore increasing network density (please cite original work).” We are not sure which “original models” the reviewer is thinking of, but as noted above, if the nucleation and capping rates remain constant (even if the elongation rate decreases) there will be no change in the density of free barbed ends. And (2) “…the authors [should] take the analysis one step further and provide, based on their results, an answer to the question they address: 'how load forces increase the number of growing filaments’.” We do, in fact, provide an explanation, which is simply this: applied forces cause a decrease in the rate at which individual filaments are capped (described by Brownian Ratchet theory) and this decrease produces the observed increase in the steady-state barbed end density as described in Eq. 2 above.

3) Related to point 1), it is true that experiments performed at different concentration of CP can only be compared if the pool of protein has not had time to evolve differently between the time when the proteins are mixed and the time when the measurements are made. Please reply to this comment (Reviewer 1 comment 3).

The concerns about time evolution of the system and nominal concentrations are most clearly expressed by Reviewer 1: “From the moment when proteins are mixed to the moment when measurements are made, how does the protein system evolve from the initial state (which is if I understand correctly 5uM G-actin, 5uM profilin, 100nM CP and 100nM Arp2/3)? At this high G-actin concentration and even with equimolar profilin, I expect that a non-negligible fraction of actin will have polymerized within seconds/minutes, and a fraction of CP and Arp2/3 will be bound to F-actin in the bulk.” In response to this comment we rewrote the manuscript to more clearly describe the metastable nature of the soluble protein pool. As the reviewer’s comment implies, profilin damps spontaneous nucleation of actin but does not significantly perturb the critical concentration for filament assembly. The key feature of our reaction mixture, however, is that it contains *both* profilin *and* capping protein, which work together to effectively suppress filament assembly. Spontaneously nucleated filaments are rapidly capped at their barbed ends. Profilin then effectively prevents elongation from the pointed ends of these filaments and they disassemble. We describe these effects in the Results section of the revised manuscript and cite previous work that establishes and discusses these synergistic activities. Briefly, Young et al. (1990) found that factors that cap >90% of filament barbed ends increase the critical concentration from that of the barbed end to that of the pointed end. Pollard and Cooper (1984) demonstrated the profilin blocks pointed end elongation, and several groups demonstrated that profilin and barbed-end capping proteins work together to suppress filament assembly and promote disassembly of filaments with free pointed ends (e.g. DeNubile, 1985; Blanchoin, 2000; Pernier, 2016). This combination produces a large pool of monomeric actin that is capable of transiently elongating any newly formed barbed ends. We previously described this pool as ‘metastable’ (Pollard, 2000) while others have described it as ‘dynamically stable’ (Pernier, 2016). Only the branched actin networks formed by the micro-patterned nucleation promoting factors have an appreciable lifetime and consume a significant fraction of the soluble proteins. This is because, at the coverslip surface, filaments are continually formed by Arp2/3-dependent nucleation and they are stabilized because their pointed ends are capped by the Arp2/3 complex (Blanchoin, 2000). In addition, the total amount of protein incorporated into the micro-patterned branched networks is only a small fraction of the total protein present in the reaction mix. This is demonstrated by the fact that the network growth rate is constant over the course of each experiment. We mention this fact in the revised manuscript and provide the following simple calculation to emphasize this point: The concentration of actin in our reaction mixes is 5 µM, with a total volume of 150 µl. The maximum concentration of actin in our networks is 1.25 mM, but the maximum total volume of these networks is only <0.002 µl (based on a total of 400 WAVE1 patches with an average area of 50 µm^2^, generating networks with a maximum height of <100 µm). The fraction of actin used up during an experiment, therefore, is less than 0.3%.

4) Regarding diffusion vs. ratchet to explain the effect of size on CP, could you provide a good argument or an easy experiment to justify that 'bulky-CP' diffuses at the same rate through branched networks as regular CP does? (Reviewer 1 Main comment 4)

Two pieces of evidence argue that diffusion does not affect the per-filament capping rate of the ‘bulky capping protein’ mutant. Firstly, the calculated mesh size of the actin network, even at the highest densities, is large compared to the size of the wild type and bulky capping proteins as well as the Arp2/3 complex (which also has no trouble diffusing to the center of the pattern). From the lowest to the highest load forces we used in our experiments the of actin filament density in the network increases from an equivalent local concentration of 125 µM to 1.25 mM. The corresponding mesh size (ζ) of the network decreases from 135 nm to 45 nm (*Schmidt, 1989). In comparison, the Stokes radii of wild type and ‘bulky’ capping proteins are 3.8 nm (Cooper, 1984) and ~5.4 nm, while the Stokes radius of the Arp2/3 complex is 5.3 nm (Mullins, 1997). All three values are well below the threshold value of 18 nm (0.4xz ; Wong, 2004), where interaction with the network begins to strongly affect diffusion. Secondly, the line scans of fluorescence intensity across the growing surface show no evidence of a ‘hole’ in the center of the network as would be expected for anomalous or restricted diffusion. There is, however, a thin, bright lip of bulky CP at the edge of the pattern, but this is likely due to a sharp fall-off in force at the edges of the pattern.

* The equation relating mesh size to local actin concentration is ζ = 1.47/sqrt(Ca), where Ca is the actin concentration in µM and ζ is the mesh size in µm (Schmidt, 1989).

References

Blanchoin L, Pollard TD, Mullins RD. (2000) Interactions of ADF/cofilin, Arp2/3 complex, capping protein and profilin in remodeling of branched actin filament networks. Curr Biol. 10(20):1273-82.

Cooper JA, Blum JD, Pollard TD. (1984) Acanthamoeba castellanii capping protein: properties, mechanism of action, immunologic cross-reactivity, and localization. J Cell Biol. 99(1):217-25.

DeNubile MJ, Southwick FS. (1985)Effects of Macrophage Profilin on Actin in the Presence and Absence of Acumentin and Gelsolin J. Biol. Chem. 260(12):7402-7409.

Mullins RD, Stafford WF, Pollard TD. (1997) Structure, subunit topology, and actin-binding activity of the Arp2/3 complex from Acanthamoeba. J Cell Biol. 136(2):331-43.

Pernier J, Shekhar S, Jegou A, Guichard B, Carlier MF. (2016) Profilin Interaction with Actin Filament Barbed End Controls Dynamic Instability, Capping, Branching, and Motility. Dev Cell. 36(2):201-14.

Pollard TD, Blanchoin L, Mullins RD. (2000) Molecular mechanisms controlling actin filament dynamics in nonmuscle cells. Annu Rev Biophys Biomol Struct. 29:545-76.

Pollard TD, Cooper JA. (1984) Quantitative analysis of the effect of Acanthamoeba profilin on actin filament nucleation and elongation. Biochemistry. 23(26):6631-41.

Schmidt (1989) Chain Dynamics, Mesh Size, and Diffusive Transport in Networks of Polymerized Actin. A Quasielastic Light Scattering and Microfluorescence Study. Macromolecules. 22:3638-3649

Wong IY, Gardel ML, Reichman DR, Weeks ER, Valentine MT, Bausch AR, Weitz DA (2004) Anomalous Diffusion Probes Microstructure Dynamics of Entangled F-Actin Networks. Phys. Rev. Lett. 92(17):17801.

Young CL, Southwick FS, Weber A. (1990) Kinetics of the Interaction of a 41-Kilodalton Macrophage Capping Protein with Actin: Promotion of Nucleation during Prolongation of the Lag Period. Biochemistry, 29:2232-2240.

All three reviewers have other interesting suggestions, those particularly deserving of your attention being:- Please cite Wiesner et al. JCB 2003. We agree that their interpretation of comet density change under force was different than yours, but it was nevertheless the first observation of this phenomenon.

We agree with the interpretation provided by Wiesner et al. in their paper, namely that the observed effect of methylcellulose on branched actin network density is a chemical effect (likely due to changes in water activity or ‘depletion interaction’ effects) and does not reflect a response to applied force. Most convincingly, the authors note that the methylcellulose-dependent density changes occurred “…with methylcellulose of different chain length; hence, it was not related to the increase in viscosity.” Moreover, the highest concentration of methyl-cellulose used in their experiments —which increased the actin network density by a factor of 4— had a minimal effect on the network growth rate (Figure 4C in Wiesner et al., 2003). From their estimates of viscous drag, the maximum force applied to their beads was ~50 pN, or roughly 15 pN/µm^2^. In our experiments a factor of four change in network density (as observed by Wiesner et al.) required an applied load of 500 pN/µm^2^, a force that slowed network growth rate by more than 80%. Wiesner et al. (2003) is an important paper, and we have included a citation in the revised manuscript. However, citing Wiesner et al. as early evidence for force-induced changes in actin network density would be both misleading and a disservice to the excellent experimental work and careful analysis presented in that paper.

This paper also nicely showed that branch density increases when capping is increased.

This is a bit misleading. The authors of that paper note that the ratio of Arp2/3 and actin fluorescence intensities (I_R_/I_A_) increases with increasing capping protein. This is equivalent to saying that the average filament length (proportional to I_A_/I_R_) decreases with added capping protein. As we note in this manuscript and others, this can also be observed by measuring the ratio of actin to capping protein in the network. Although not noted in the text, this observation likely makes the Wiesner paper the first experimental demonstration of the effect of capping protein on average filament length in a branched actin network, and we will certainly cite it. The I_R_/I_A_ ratio, however, does not reflect branch density (at not least in the sense of branches per µm^3^), because most (or all) of the change in that ratio comes from a decrease in I_A_ rather than an increase in I_R_ (see, e.g., Figures 1B and 1D in Akin and Mullins, 2008). To see why I_R_/I_A_ does not reflect branch density see Figure 3D-E, from Bieling et al., 2016. Here, applied force induces large changes in branch density —judged by the absolute Arp2/3 intensity (not shown) and the surface density of growing barbed ends (top panel)— with little or no change in average filament length, hence no change in the ratios of Arp2/3 or capping protein to actin (bottom panel).

Changes in branch density can only be judged by counting single molecules or comparing absolute intensities of the Arp2/3 complex incorporated into networks under various conditions. Similarly, the effect of capping protein on the overall rate of branch formation can only be observed in this system by multiplying the absolute Arp2/3 intensity by the rate of network growth (which Wiesner et al. also measure but do not apply in this way).

- Interesting comments from Reviewer 2 (second and third recommendation) and 3 (whether the conclusions are universal for all Arp2/3 complex activators) to expand the discussion. Experiments with other Arp2/3 activators and actin regulators would be great, but represent too much work for this paper.

These differences are discussed in more detail in the revised manuscript, and we provide additional data (e.g. line scans of component concentrations) and calculations addressing the possible effects of differential diffusion.

- Comments of Reviewer 3 about data representation and analysis (justification of Student's t tests; data for Arp2/3 dwell times at 25 Pa; comment about plot shape in Figure 2D and 2F)

These were helpful comments, and we revised our analysis and presentation in response to them. We initially used a Student’s t-test to calculate p-values, in part because it is quite robust to differences in variance (Ramsey, 1980), but the reviewer is correct to point out that this is not appropriate. We have, therefore, recalculated p-values for all the relevant data using a paired Wilcoxon signed rank test, which does not assume that differences between paired samples are normally distributed (Siegel, 1956). We also changed the presentation of our histograms in the new versions of Figures 3D and 3F (2D and 2F in the original manuscript).

- Reviewer 2 and 3 both thought that Figure Supplement 1 should be part of a main Figure.

We have made the suggested change to the revised manuscript.

References

Akin O, Mullins RD. (2008) Capping protein increases the rate of actin-based motility by promoting filament nucleation by the Arp2/3 complex. *Cell*. 133(5):841-851.

Ramsey PH. (1980) Exact type 1 error rates for robustness of Student’s t test with unequal variances. *J. Educational and Behavioral Statistics*. 5(4):337-349.

Siegel S. (1956) Non-parametric statistics for the behavioral sciences. New York: McGraw-Hill. pp. 75–83. ISBN 9780070573482.

Wiesner S, Helfer E, Didry D, Ducouret G, Lafuma F, Carlier MF, Pantaloni D. (2003) A biomimetic motility assay provides insight into the mechanism of actin-based motility. *J Cell Biol.* 160(3):387-98.